# Observation of dichotomic field-tunable electronic structure in twisted monolayer-bilayer graphene

Hongyun Zhang [1,12], Qian Li [1,12], Youngju Park [2], Yujin Jia[3,4], Wanying Chen[1], Jiaheng Li[3,4], Qinxin Liu[1], Changhua Bao[1], Nicolas Leconte [2], Shaohua Zhou [1], Yuan Wang[1], Kenji Watanabe [5], Takashi Taniguchi [6], Jose Avila [7], Pavel Dudin [7], Pu Yu [1,8], Hongming Weng [3,4,9], Wenhui Duan [1,8,10], Quansheng Wu [3,4], Jeil Jung [2,11] & Shuyun Zhou [1,8] ✉

Twisted bilayer graphene (tBLG) provides a fascinating platform for engineering flat bands and inducing correlated phenomena. By designing the stacking architecture of graphene layers, twisted multilayer graphene can exhibit different symmetries with rich tunability. For example, in twisted monolayer-bilayer graphene (tMBG) which breaks the $C_{2z}$ symmetry, transport measurements reveal an asymmetric phase diagram under an out-of-plane electric field, exhibiting correlated insulating state and ferromagnetic state respectively when reversing the field direction. Revealing how the electronic structure evolves with electric field is critical for providing a better understanding of such asymmetric field-tunable properties. Here we report the experimental observation of field-tunable dichotomic electronic structure of tMBG by nanospot angle-resolved photoemission spectroscopy (NanoARPES) with operando gating. Interestingly, selective enhancement of the relative spectral weight contributions from monolayer and bilayer graphene is observed when switching the polarity of the bias voltage. Combining experimental results with theoretical calculations, the origin of such field-tunable electronic structure, resembling either tBLG or twisted double-bilayer graphene (tDBG), is attributed to the selectively enhanced contribution from different stacking graphene layers with a strong electron-hole asymmetry. Our work provides electronic structure insights for understanding the rich field-tunable physics of tMBG.

Magic angle twisted bilayer graphene (tBLG) has attracted extensive research interests due to the flat band[1] near the Fermi energy $E_F$ with emergent correlated phenomena, such as superconductivity[2], Mott insulating state[3], and ferromagnetism[4,5]. By increasing the number of graphene layers, twisted multilayer graphene (tMLG) can exhibit a rich spectrum of stacking configurations with distinct symmetries[6–9], providing additional controlling knobs for tailoring the physical properties. For example, tunable spin-polarized correlated states have been reported in twisted double-bilayer graphene (tDBG)[10–13], and correlated states with non-trivial band topology have been reported in rhombohedral trilayer graphene[14–16].

Among various structures of tMLG, twisted monolayer-bilayer graphene (tMBG) with a lower symmetry is particularly fascinating[17–22]. Unlike tBLG and tDBG which have symmetric stacking, the asymmetric

stacking of monolayer graphene on Bernal-stacked bilayer graphene in tMBG breaks the $C_{2z}$ symmetry (Fig. 1a), making it asymmetric or "polar" along the out-of-plane direction. Applying an out-of-plane electric field can further enhance the asymmetry, giving rise to a rich phase diagram which strongly depends on the field direction. So far, transport measurements have reported an asymmetric field-tunable phase diagram[19], which changes from a correlated phase (similar to tBLG) to a ferromagnetic phase (reminiscent of tDBG) when reversing the electric field direction. Such field-tunable correlated phenomena suggest a strong modification of electronic structure under the application of an electric field. Directly probing how the actual electronic structure evolves with electric field is therefore critical for providing a better understanding of such dichotomic field-tunable physics.

Here, by performing NanoARPES measurements (see schematic illustration in Fig. 1a) on tMBG with a twist angle of 2.2° with operando gating, we report the observation of a dichotomic response of the electronic structure to the electric field (induced by a bias voltage). The main experimental results are schematically summarized in Fig. 1b–d and supported by data shown in Fig. 1e–g. We find that, although the bottom bilayer graphene (2 ML) always has a weaker spectral intensity compared with top monolayer (1 ML) graphene, a positive bias voltage (induced electric field $E_{ind}$ pointing from monolayer to bilayer) enhances the relative spectral weight contribution from 1 ML graphene (pointed by red arrow in Fig. 1e) as well as the conical shaped feature, while a negative bias voltage enhances the relative contribution from 2 ML graphene, leading to flatter electronic structure near the Fermi energy (Fig. 1g). A comparison between the experimental results with theoretical calculations allows to reveal the origin of dichotomic field-tunable properties of tMBG from an electronic structure perspective.

## Identifying the flat band, monolayer and bilayer graphene features in tMBG

The high-quality tMBG sample was prepared by the clean dry transfer method[23,24] (see method and Supplementary Figs. 1, 2 for more details). The twist angle was determined from the moiré period using lateral force atomic force microscope (L-AFM) measurements and further confirmed by NanoARPES measurements (see Supplementary Figs. 3, 4). Figure 2a–f shows NanoARPES intensity maps measured at energies from $E_F$ to −0.5 eV with twist angle of 2.2°. The Fermi surface map shows two intensity spots at the Brillouin zone (BZ) corners of the top monolayer graphene (red dot, $K_1$) and the bottom bilayer graphene (blue dot, $K_2$) respectively, with weaker replicas at the moiré superlattice Brillouin zone (mSBZ) corners. Moving down in energy, these spots expand into pockets and hybridize with each other, resulting in a flower-shaped pattern at higher binding energy (Fig. 2d–f). Here the choice of twist angle of 2.2° is ideal for the investigation of the electronic structure and its field tunability, because the slightly larger angle than the magic angle makes it easier to resolve the contributions from monolayer and bilayer graphene, meanwhile the flat band can still be observed.

The high-quality NanoARPES data allow to resolve the flat band and identify spectral features from monolayer and bilayer graphene. Figure 2g shows dispersion image measured by cutting through two mSBZ corners as schematically illustrated in the inset. An isolated flat band (pointed by red arrow) is observed with a clear hybridization gap (pointed by black arrow), which separates the flat band from the remote bands at higher binding energy. Similar features are also captured by the calculated spectrum shown in Fig. 2h using an effective tight-binding model (see Method for more details), where red and blue colors represent projected contributions from the top 1 ML and bottom 2 ML graphene layers respectively. From the experimental data, the extracted bandwidth of the flat band is 70 ± 10 meV (see Supplementary Fig. 5 for more details), which is comparable to the calculated Coulomb energy[3], $U_{eff} = e^2/(4\pi\epsilon_0\epsilon_r\lambda_m) = 75$ meV, where $\epsilon_0$ and $\epsilon_r$ are the dielectric constants of the vacuum and the substrate respectively, and $\lambda_m$ is the moiré period. The similar energy scales between the bandwidth and the Coulomb energy suggest that 2.2° tMBG is near the correlated regime, in line with the larger range of twist angle where the flat band exists[9,25,26]. Figure 2i shows dispersion image measured by cutting through the BZ corners of both monolayer and bilayer graphene, and the calculated spectrum is shown in Fig. 2j for comparison. An overall conical dispersion (guided by red dashed curves in Fig. 2i) is observed around $K_1$ together with their moiré replica bands (brown

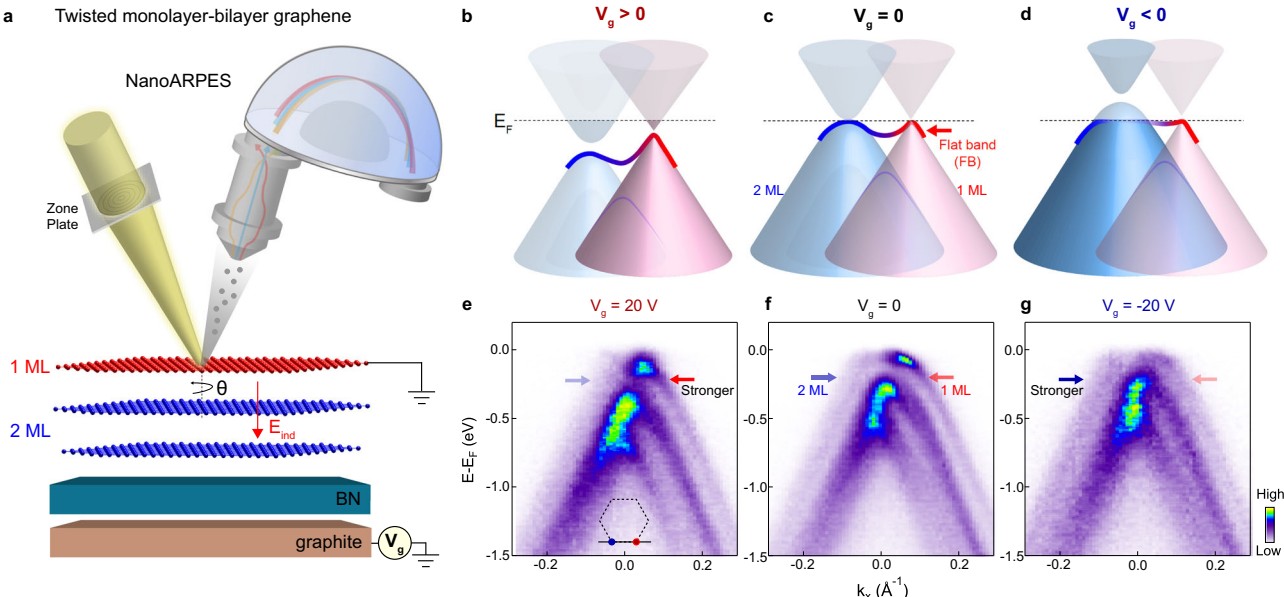

**Fig. 1 | Schematic summary of the electric field-tunable electronic structure in tMBG. a** Schematic for NanoARPES measurements of tMBG with operando gating capability. The red arrow indicates the induced electric field $E_{ind}$ among three graphene layers under positive bias voltage. **b**–**d** Schematic summary of the field-tunable electronic structure. **e**–**g** Dispersion images measured under positive, zero, and negative bias voltages, respectively. Colors represent the NanoARPES measured intensity as indicated by the colorbar. The measurement direction is that connecting the BZ corners of the top (red dot, $K_1$) and bottom (blue dot, $K_2$) graphene, as indicated by the black line in the inset of (**e**).

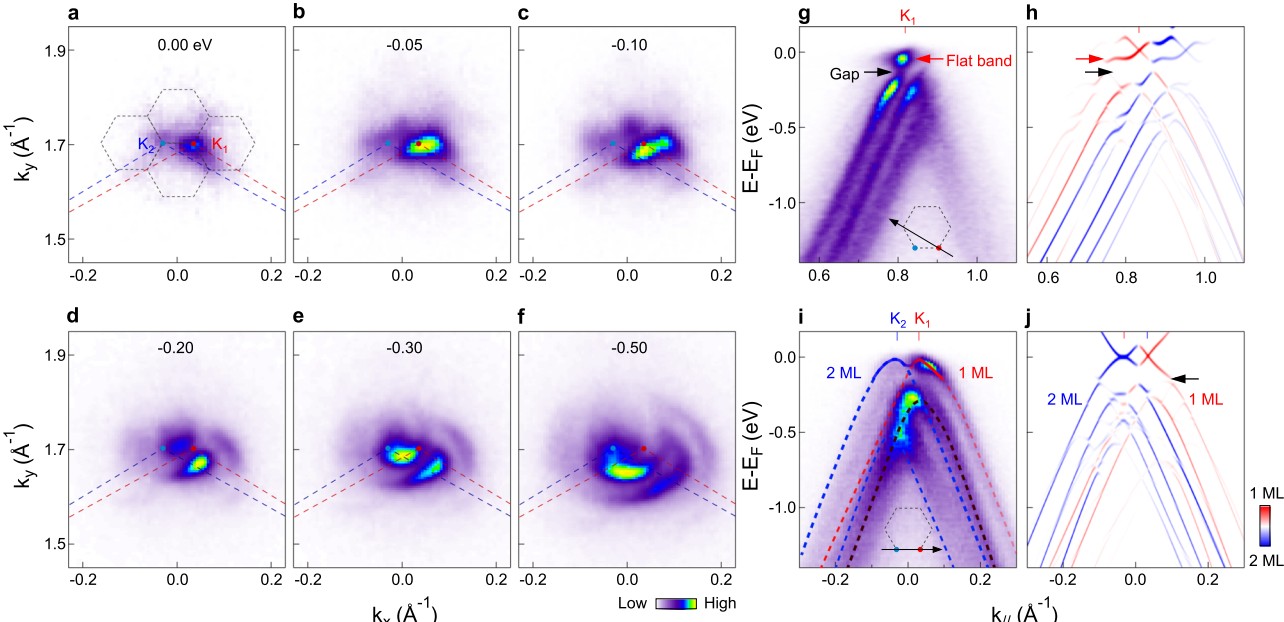

**Fig. 2 | Fermi surface topology and coexistence of monolayer and bilayer graphene features in the 2.2° tMBG. a–f** ARPES intensity maps measured at energies from $E_F$ to −0.50 eV. The red and blue lines mark the Brillouin zone boundaries for top monolayer graphene and bottom bilayer graphene respectively. The black dotted hexagons represent the moiré Brillouin zones. **g, h** Dispersion image measured along the black line indicated by the inset of (**g**), and calculated spectrum for comparison. Red and blue colors in (**h**) represent projected contributions from the top 1 ML and bottom 2 ML graphene layers, respectively. Red and black arrows point to the flat band and hybridization gap respectively. **i, j** Dispersion image measured along the black line indicated by the inset of (**i**), and calculated spectrum for comparison. Red, blue and brown dashed curves indicate the bands from 1 ML, 2 ML and the moiré replica band.

dashed curves), and two parabolic bands from bilayer graphene (blue dashed curves) are observed near $K_2$. In the region where these bands overlap, there is a strong intensity suppression and the flat band emerges as a result of the interlayer interaction (see Supplementary Fig. 5), similar to the case of tBLG[27,28], while here the asymmetric stacking of tMBG provides additional field-tunability. We note that while the hybridization gap (pointed by black arrow in Fig. 2j) is too small to be resolved from the experimental data directly, however, a sudden change of intensity is observed at the flat band edge (Fig. 2i), suggesting an overall agreement with the calculated spectrum in Fig. 2j. The comparison between experimental results and theoretical calculations allows to reveal the flat band as well as spectroscopic contributions from monolayer and bilayer graphene, which lays an important foundation for further investigating the field-tunable electronic structure.

## Dichotomic field-tunable electronic structure

Figure 3a–j shows an overview of dispersion images measured through $K_1$ and $K_2$ with bias voltages from −20 V to 30 V. Here the bias voltage $V_g$ is applied on the bottom gate, which not only tunes the carrier concentration $n$, but also applies an electric field[29,30]. There are a few observations from the evolution of the electronic structure. First of all, a negative bias voltage ($E_{ind}$ pointing from bilayer to monolayer graphene) dopes the sample with holes and shifts the bands up, while a positive bias voltage leads to electron doping and shifts the bands down, showing the same trend as gated graphene devices[29–32]. From 0 V to 30 V, the bands shift in energy by 140 ± 30 meV (see Supplementary Fig. 6), which is consistent with the estimated carrier density from the geometric capacitance (see Supplementary Note 3 for more details). Secondly, the flat band near the Fermi energy can be selectively tuned to be more extended (flatter) or dispersive by switching the bias voltage (see Supplementary Fig. 7 for more details). For negative bias voltage, the flat band becomes more extended in the momentum space (red dotted curve in Fig. 3a), while for positive bias voltage, more pronounced "M-shaped" dispersion with a conical

behavior is observed (red-to-blue dotted curve in Fig. 3j). Thirdly, the relative spectral weight contributions from monolayer and bilayer graphene can also be selectively enhanced by reversing the bias voltage, as indicated by red arrow in Fig. 3j and blue arrow in Fig. 3a (see Supplementary Fig. 8). This is also evident in the momentum distribution curves (MDCs) measured at a few representative bias voltages in Fig. 3k, where the red and blue dashed arrows indicate the relative spectral weight transfer to monolayer and bilayer graphene under positive and negative bias voltages respectively. Figure 3q shows a comparison between MDCs measured at bias voltages of −20 V (blue curve) and 30 V (red curve), where a relative spectral weight transfer from 2 ML to 1 ML bands is clearly observed when increasing the bias voltage.

The dichotomic field response of the electronic structure is also revealed in the calculated spectra in Fig. 3l–p, where a stronger relative spectral weight contribution is observed for monolayer valence band at positive bias voltage. Interestingly, at negative bias voltage (Fig. 3l), the enhanced relative spectral weight contribution from bilayer graphene in the valence band (blue arrow in Fig. 3l) is also accompanied by a reduced spectral weight contribution from the conduction of bilayer graphene (light blue arrow in Fig. 3l), suggesting a stronger electron-hole asymmetry than that at zero bias voltage (Fig. 3n). In addition, the application of a bias voltage also leads to flatter dispersion for bilayer graphene bands, similar to gated bilayer graphene[33]. Moreover, the calculated spectra show that the valence band at positive bias voltage shows similar dispersion to the conduction band at negative bias voltage (see Supplementary Fig. 9 for more details), suggesting that reversing the bias voltage can lead to a switching between the electron and hole bands in tMBG.

## Origin of the field-tunable dichotomic electronic structure

To reveal the origin of the spectral weight transfer and the field-tunable electronic structure of tMBG, we show in Fig. 4 the comparison of calculated energy contours at −0.2 eV for tBLG, tMBG and tDBG

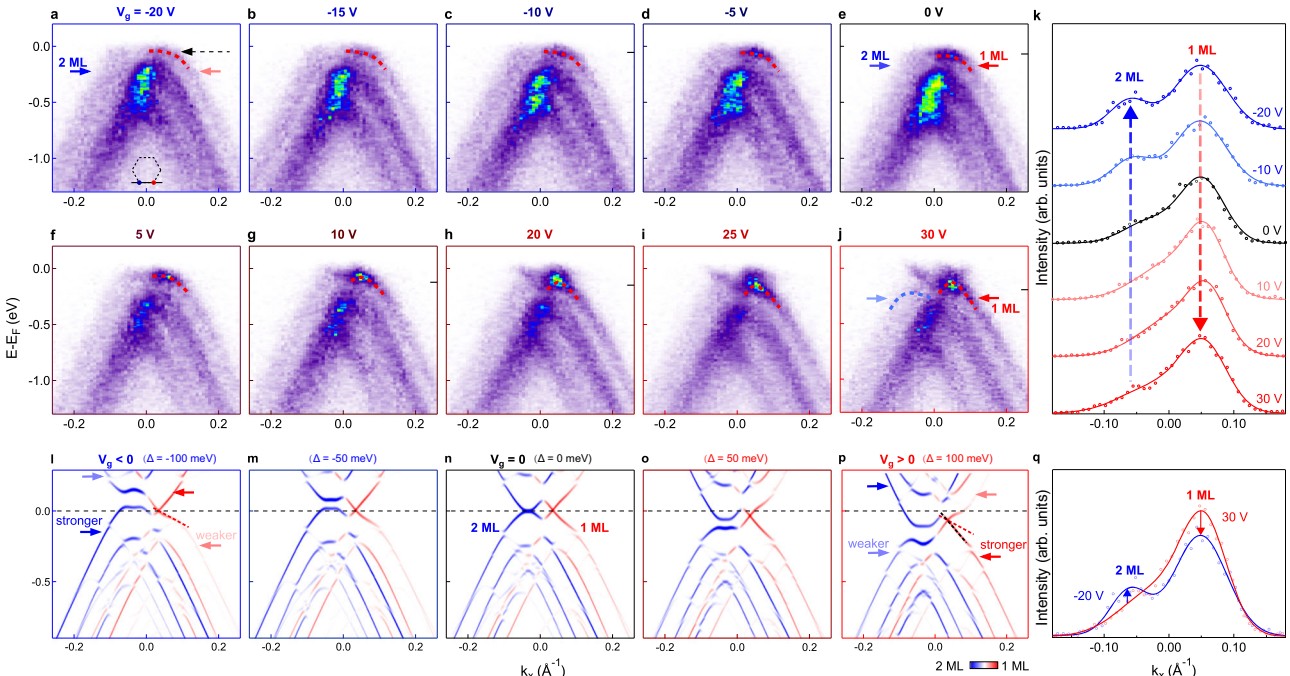

**Fig. 3 | Dichotomic field response of the electronic structure under negative and positive bias voltages in a 2.2° tMBG. a–j** Dispersion images measured through $K_1$ and $K_2$ (indicated by the inset) with bias voltages from -20 V to 30 V. Dotted red and blue curves in (**a–j**) are guiding curves. **k** Representative MDCs extracted at energies indicated by gray dashed arrow in (**a**) and black tick marks in (**c, e, g, h, j**) to reveal the selectively enhanced spectral weight from 1 ML (pointed by red dashed arrow) and 2 ML graphene (pointed by blue dashed arrow) under positive and negative bias voltages, respectively. Colored marks and curves

correspond to the experimental results and fitting curves. **l–p** Calculated spectra at negative (**l, m**), zero (**n**), and positive (**o, p**) bias voltages, with red and blue colors representing the contribution from 1 ML and 2 ML graphene respectively. The values of the interlayer potential difference (Δ) used for the calculations are Δ = −100, −50, 0, 50 and 100 meV respectively. **q** Comparison of MDCs extracted at −20 V and 30 V, which shows relatively enhanced 2 ML or 1 ML bands at negative and positive bias voltages.

under positive and negative bias voltages, respectively. For tBLG and tDBG, the energy contours remain similar when reversing the bias voltage (see comparison between Fig. 4a, d and Fig. 4c, f), except that the pattern centering at $K_1$ now switches to $K_2$ and vice versa, which is consistent with the overall symmetric phase diagram from transport measurements of tBLG[4,34] and tDBG[11–13]. In sharp contrast, reversing the bias voltage leads to a dramatic change in the energy contour of tMBG (see Fig. 4b, e). Remarkably, the energy contour of tMBG under positive bias voltage (Fig. 4b) is strikingly similar to that for tBLG (Fig. 4a), while the energy contour under negative bias voltage (Fig. 4e) resembles that of tDBG (Fig. 4f), again supporting the asymmetric or "polar" electric field response.

To resolve the puzzle of how tMBG under bias voltage can exhibit electronic structures similar to tBLG and tDBG, we show in Fig. 4g–l energy contours at −0.2 eV projected onto each constitute graphene layers for tBLG, tMBG, and tDBG. For these three different types of twisted structures, the energy contours for graphene layers above the interface ($L_3$ and $L_4$) all exhibit clockwise vortex pattern centered at $K_1$ (red dot), while the bottom layers ($L_1$ and $L_2$) show counter-clockwise vortex pattern centered at $K_2$ (blue dot). This is consistent with their relative rotation directions in the real space, and reflects the chiral properties of twisted graphene structures[35,36]. Moreover, for positive bias voltage, the energy contours for the top graphene layers ($L_3$ and $L_4$) show a larger pocket size with an enhanced spectral weight, suggesting the enhanced contributions from top graphene layers (Fig. 4g–i), while for negative bias voltage, contributions from the bottom layers are enhanced (Fig. 4j–l) (see Supplementary Fig. 10 for the calculated layer-resolved density of states (DOS)). Therefore, although tMBG has one more layer (bottom layer, $L_1$) than tBLG, the energy contour for tMBG under positive bias voltage is still similar to that of tBLG due to the smaller contribution of spectral weight from $L_1$. The energy contour for tDBG, however, is quite different from tMBG, because tMBG lacks the

top layer $L_4$, which has a strong spectral weight. Similarly, for negative bias voltage, the energy contour of tMBG is overall similar to that of tDBG due to the smaller pocket of $L_4$. Although the modulation in the spectral weight contribution of different layers is small, the asymmetric shape of pockets under reversed bias voltage is still significant enough to explain the origin of the overall asymmetric behavior in the tMBG.

The layer-projected electronic structure analysis suggests that the dichotomic field-tunable electronic structure under positive and negative bias voltages originates from the selectively enhanced contributions from different constitute graphene layers, which is also intrinsically related to the breaking of the $C_{2z}$ symmetry in tMBG. While a field-tunable electronic structure has been deduced from the asymmetric phase diagram reported in transport measurements[19], our work provides direct electronic structure insights for understanding the field-tunable physics. In particular, by projecting the spectral contribution from each individual layer in the momentum space, our work also provides more complete and microscopic information on how the electric field actually tunes the electronic structure of each individual layer. We envision that similar strategies can be extended to other asymmetric twisted bilayer or multilayer systems, where the crystalline symmetry can be used as a tuning knob to obtain exotic properties, such as gate-tunable ferroelectricity and nonlinear optical response.

## Methods
### Sample preparation with gating capability
The tMBG sample was prepared by using the clean dry transfer method[23,24]. First, the graphene flake with both monolayer and bilayer parts connected together was exfoliated onto a clean SiO$_2$/Si substrate. A thin BN flake attached to PVA/PDMS (Polyvinyl Alcohol/Poly-dimethylsiloxane) was then positioned above the graphene under an optical microscope to pick up the bilayer graphene part. The monolayer graphene on the SiO$_2$/Si substrate was rotated by the desired

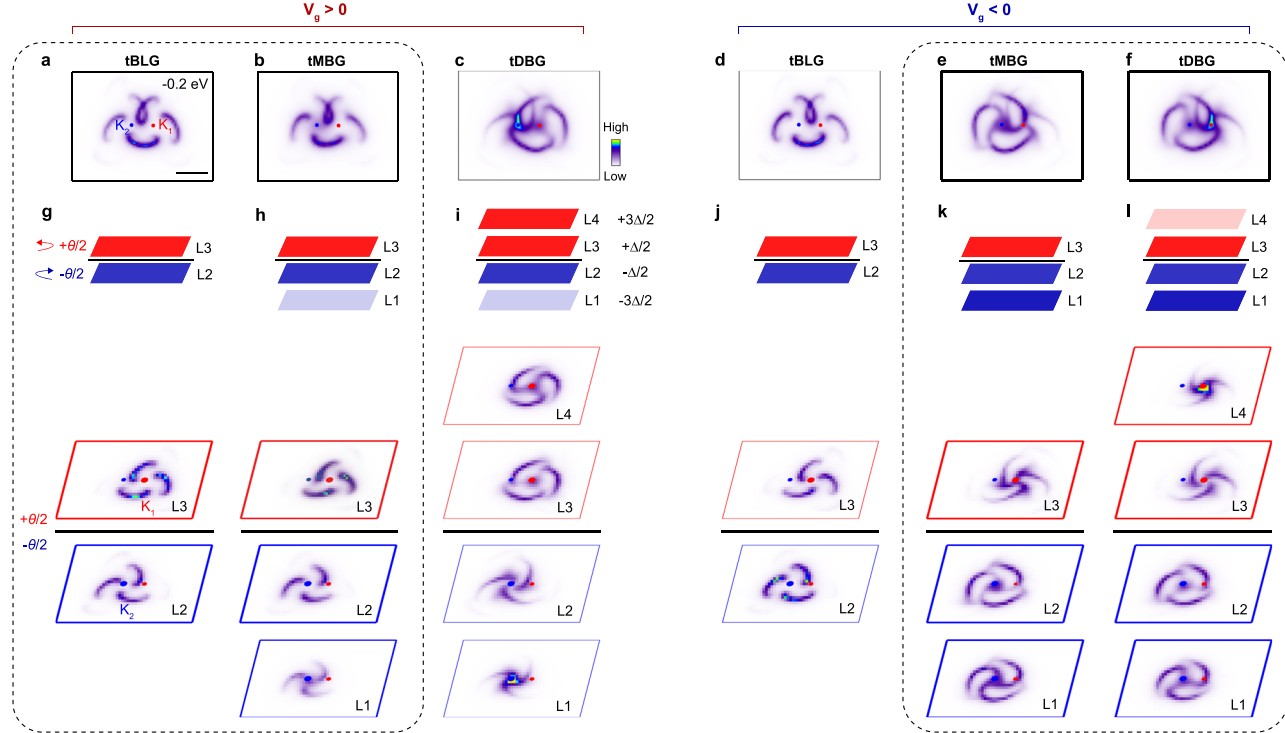

**Fig. 4 | Field-tunable electronic structure from theoretical calculations, and layer-resolved energy contours for positive and negative bias voltages.**
**a–f** Calculated intensity maps at constant energy of -0.2 eV for tBLG, tMBG and tDBG under positive (**a–c**) and negative (**d–f**) bias voltages with interlayer potential difference of Δ = 100 meV and −100 meV respectively. The scale bar in (**a**) is 0.1 Å⁻¹.
**g–i** Schematic illustrations of field-induced selectively enhanced contribution under positive bias voltage from different layers of tBLG, tMBG and tDBG (indicated by dark and light shaded colors in the top panel schematic), and calculated layer-

projected energy contours at −0.2 eV (lower panels, positive bias voltage with Δ = 100 meV). **j–l** Schematic illustrations of field-induced selectively enhanced contribution from different layers of tBLG, tMBG and tDBG under a negative bias voltage (top panels), and calculated layer-projected intensity maps at −0.2 eV (lower panels, Δ = −100 meV). The dotted rectangles are used to highlight the similarity between tBLG and tMBG at positive bias voltage, and tMBG and tDBG at negative bias voltage.

angle and picked up by bilayer-graphene/BN/PVA/PDMS to form the tMBG/BN/PVA/PDMS structure. Subsequently, the tMBG/BN/PVA/PDMS was flipped over, and the tMBG/BN/PVA was picked up by another PDMS stamp to form PVA/BN/tMBG/PDMS structure. The PVA film was dissolved by immersing the entire structure in water for several hours, and the tMBG/BN was transferred onto a graphite flake, which was in contact with the gold-plated pattern as the bottom gate electrode. Finally, two narrow pieces of graphite were used to electrically connect tMBG and the gold-coated pattern to ensure good electrical conductivity for ARPES measurements. The gold-plated pattern was connected to the gating electrode of the sample holder by wire bonding. See Supplementary Figs. 1, 2 for more details.

## AFM measurements

The twist angle can be determined by combining NanoARPES and lateral force AFM (L-AFM) measurements. For the L-AFM measurements, silicon nitride probes were used to obtain the lateral force and topography image under the contact mode. We note that L-AFM measurement is particularly sensitive to the moiré superlattice period because the stick-slip effect would occur at the moiré superlattice scale[37]. After extracting the moiré superlattice period $\lambda_m$ from L-AFM measurements, the twist angle $\theta$ can be further determined by $\lambda_m = a/(2sin(\theta/2))$.

## ARPES measurements

The tMBG sample was annealed at 150 °C in ultrahigh vacuum (UHV) until sharp dispersions were observed. NanoARPES measurements were performed at the beamline ANTARES of the synchrotron SOLEIL in France with a beam size of 500–700 nm and photon energy of 100 eV. The incident light is set to $p$-polarized and the analyzer slit direction is

horizontal (see Supplementary Fig. 12). The overall energy and angular resolution were set to 50 meV and 0.1°, respectively. The measurement temperature was 70 K in a working vacuum better than $2 \times 10^{-10}$ mbar.

## Theoretical calculations

The electronic structure calculations are performed by using a real-space tight-binding model, where we define the Hamiltonian for tBLG, tMBG, and tDBG as

$$H = \sum_{i,j} t_{ij} c_i^\dagger c_j + \sum_i (\varepsilon_i + V_{\ell \in i}) c_i^\dagger c_i \qquad (1)$$

where $\varepsilon_i$ and $t_{ij}$ are the on-site potential energy of atom $i$ and the hopping parameter between atom $i$ and $j$. We use the effective tight-binding model parameters for $\varepsilon_i$ and $t_{ij}$ using either the monolayer[38] version or the Bernal bilayer[39] version depending on which part of the system is under consideration. We set the effective nearest hopping term $t_0 = -3.1$ eV and the interlayer hopping scaling factor S = 0.895 for the Scaled Hybrid Exponential (SHE) model[40], to effectively match the Fermi velocity $v_F \approx 1 \times 10^6$ m/s and the magic angle 1.08° of tBLG. In the calculations, a commensurate superlattice is used, and here we use a twist angle of 2.28° to perform the calculations, which is the closest commensurate twist angle near 2.2°. Classical structural relaxation using LAMMPS[41,42] has been implemented in the calculation. In order to describe the electric field effect under applying bias voltages, we label the graphene layers from $L_1$ (bottom layer) to $L_4$ (top layer) as indicated by Fig. 4, and the potential energy for different layers are set to be $V_1 = -V_4 = -3\Delta/2$ and $V_2 = -V_3 = -\Delta/2$. We note that a positive bias voltage $V_g > 0$ gives rise to $\Delta > 0$, leading to an induced electric field

$E_{ind}$ pointing from monolayer to bilayer graphene, where the total electric field contains $E_{total} = E_{ext} + E_{ind}$ contains the external electric field $E_{ext}$ and the induced electric field $E_{ind}$. An adjustable chemical potential $\mu$ is used to introduce an overall shift in the tBLG, tMBG and tDBG bands to allow a fair comparison between bands at specific energy cuts (Fig. 4). To simulate the intensity of the ARPES data, we calculate the sublattice-resolved spectral functions using a band-unfolding method[43–45], which enables to obtain the layer- or sublattice-resolved electronic structure calculation. The band unfolding code can be found as WannierTools[46] on the GitHub. See Supplementary Fig. 11 and more calculation details in the Supplementary Information.

## Data availability
All relevant data of this study are available within the paper and its Supplementary Information files. Source data are provided with this paper.

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

## Acknowledgements

This work is mainly supported by the National Key R&D Program of China (Grant No. 2021YFA1400100, 2022YFA1403800), the National Natural

Science Foundation of China (Grant No. 12234011, 52025024, 92250305, 11725418, 52388201, 12274436). H.Z. and C.B. acknowledge support from the Shuimu Tsinghua Scholar program, the Project funded by China Postdoctoral Science Foundation (Grant No. 2022M721887, 2022M721886), and the National Natural Science Foundation of China (12304226). K.W. and T.T. acknowledge support from the JSPS KAKENHI (Grant Numbers 20H00354 and 23H02052) and World Premier International Research Center Initiative (WPI), MEXT, Japan. H.W. and Y.J. acknowledge support from the Informatization Plan of the Chinese Academy of Sciences (CASWX2021SF-0102). J.J. acknowledges the funding from the National Research Foundation of Korea (NRF) through grant numbers NRF2020R1A2C3009142, the support by the Korean Ministry of Land, Infrastructure and Transport (MOLIT) from the Innovative Talent Education Program for Smart Cities and the computational support from KISTI Grant No. KSC-2022-CRE-0514 and the resources of Urban Big data and AI Institute (UBAI) at UOS. Y.P. and N.L. were supported by the NRF through grant numbers NRF2020R1A5A1016518 and the Korean NRF through Grant RS-2023-00249414. We acknowledge SOLEIL for the provision of synchrotron radiation facilities.

## Author contributions

Shuyun Z. conceived the research project. H.Z., Qian L., W.C., J.A., P.D., Qinxin L., and Shuyun Z. performed the NanoARPES measurements and analyzed the data. C.B., Shaohua Z. and Y.W. contributed to the data analysis and discussions. Qian L. prepared the tMBG samples. Qian L. and P.Y. performed the AFM measurements. K.W. and T.T. grew the BN crystals. Y.P., Y.J., J.L., N.L., Q.W., H.W., W.D., and J.J. performed the calculations. H.Z., Qian L., and Shuyun Z. wrote the manuscript, and all authors contributed to the discussions and commented on the manuscript.

## Competing interests

The authors declare no competing interests.

## Additional information

[1]State Key Laboratory of Low-Dimensional Quantum Physics and Department of Physics, Tsinghua University, Beijing 100084, PR China. [2]Department of Physics, University of Seoul, Seoul 02504, Korea. [3]Beijing National Laboratory for Condensed Matter Physics and Institute of Physics, Chinese Academy of Sciences, Beijing 100190, PR China. [4]University of Chinese Academy of Sciences, Beijing 100049, PR China. [5]Research Center for Electronic and Optical Materials, National Institute for Materials Science, 1-1 Namiki, Tsukuba 305-0044, Japan. [6]Research Center for Materials Nanoarchitectonics, National Institute for Materials Science, 1-1 Namiki, Tsukuba 305-0044, Japan. [7]Synchrotron SOLEIL, L'Orme des Merisiers, Departamentale 128, 91190 Saint-Aubin, France. [8]Frontier Science Center for Quantum Information, Beijing 100084, PR China. [9]Songshan Lake Materials Laboratory, Dongguan, Guangdong 523808, PR China. [10]Institute for Advanced Study, Tsinghua University, Beijing 100084, PR China. [11]Department of Smart Cities, University of Seoul, Seoul 02504, Korea. [12]These authors contributed equally: Hongyun Zhang, Qian Li. ✉e-mail: syzhou@mail.tsinghua.edu.cn

