## [Peer Review File · Nature Communications]

REVIEWER COMMENTS

Reviewer #1 (Remarks to the Author):

In the submitted paper, Zhang and co-authors study the angle-resolved photoemission spectra of single-gated twisted monolayer-bilayer graphene (tMBG). They show that at small twist angles ~ 2 degrees, gating the system leads to spectroscopic features resembling twisted bilayer graphene (tBG) or twisted double bilayer graphene (tDBG) for positive and negative bias, respectively.

The text is well written, the ARPES data is of high quality and the theoretical modelling is useful in establishing the message of the paper. However, the tunability of tMBG and its similarity to tBG/tDBG have been noticed elsewhere [Nature Physics 17, 374 (2021)]. The novelty of the submitted paper is the theoretical analysis of the wave functions correlated with the photoemission data. Given that the referenced paper already discusses the layer-resolved electronic density of states, its behavior in the external electric field and comparison to tBG/tDBG, I do not agree with the authors that the mechanism responsible for the tunability of tMBG remains elusive and am not convinced that the momentum resolution provided by ARPES generates enough novelty on its own to warrant publication in Nature Communications. This is also because, in my opinion, the comparison between ARPES and theory was not properly exploited. More detailed comments are below:

- 1) In the abstract, “combing” should be “combining”.
- 2) Fig. 1: “the measurement direction is along the BZ corner (...)” should be probably “the measurement direction is that connecting the BZ corners (...)”
- 3) Fig. 2: in the description of panels g,h, no explanation is given of what the calculation result actually shows. Neither is it mentioned in the main text at this point that the theory shows wave function projections.
- 4) Fig. 3, panels l-p: caption states that what is shown are the electronic dispersions. However, dispersions only provide the positions of the bands – what do the colors from blue to red reflect (again, projections of the wave functions are in fact shown?)? In general, it seems that the authors use the word dispersion somewhat ambiguously. I think the experimental graphs show photoemission intensity along some directions in the reciprocal space or at constant binding energies – they do not show electronic dispersion. The distinction might be more than academic given the well-known impact of electronic pseudo-spin on the measured intensity in graphene-based materials.

5) Given that the authors discuss the changes in the spectral function highlighting charge transfer between layers and either twisted bilayer- or twisted double bilayer-like behavior, I wonder how the electron escape depth effect interplays with the reported findings.

6) Similarly to the previous comment, there is likely some impact of the electronic pseudo-spin. Taking into account this and the previous point would allow for a better test of the theoretical wave functions.

7) I found no mention of the ARPES geometry (especially light polarization used) in the Methods – overall, a short comment might be useful for specialists.

Reviewer #2 (Remarks to the Author):

This manuscript presents a study of the electronic band structure in twisted monolayer-bilayer graphene with a 2.2-degree twist angle using nano-ARPES. The device has a back gate that can change both the carrier density and electric field. By cross-checking with their band structure calculation, they assign the two branches of bands observed in measurements to mainly coming from the monolayer and bilayer graphene respectively. They also suggest that the change of the spectral weight of the two branches under different gate voltages is consistent with their calculated bands under different electric fields. Finally, they point out through band structure calculation that the bands of twisted mono-bilayer graphene resemble those of twisted bilayer graphene (tBLG) or twisted double layer graphene (tDBG) when gate voltage is such that electrons are pushed more toward the monolayer graphene or the bilayer graphene.

Upon reviewing the manuscript, I have several questions that I hope the authors can clarify:

1. Determination of k-Space Orientation (Figures 1 and 2): From the ARPES measurement one can measure the relative momentum. How did the authors determine the absolute orientation and position in k-space? It seems that the authors attributed the bright spot in the equal energy cut at $E - E_F = 0$ in Fig.2a to monolayer graphene Dirac cone and therefore it's K1, the mini-Brillouin zone (mBZ) corner. This seems reasonable. However the basis for determining K2 is not apparent because data there is a purple blob and there's no distinct features like K1. Additionally, how is k-space orientation determined as indicated by the blue and red dashed line in Fig.2 a-f? This may require looking at the bands across the larger k-space to display original Brillouin zone (BZ) of graphene and reveal hexagons of Dirac points, such as presented in supplementary Fig.4 of the paper "Nature Physics volume 17, pages184–188 (2021)". Do the authors have data in larger momentum range to show this? And it would be nice to show the hexagons for bilayer graphene too.

2. Isolated Flat Band Discussion (Page 7): The manuscript states the observation of an isolated flat band. However, although it looks isolated in Fig. 2g and h with a hybridization gap, the same bright spot along a different cut in Fig. 2i and j is not isolated, but continuously connecting to the “1 ML” dispersive band. Isn't this contradictory to having an isolated flat band?
3. Clarity of some Descriptions (Page 8): The manuscript says “For negative bias voltage, the flat band becomes more extended in the momentum space (see Fig. 3a)”. However this flat band extension is ambiguous because Fig. 3a only shows that the bright spot disappeared and there's no clear flat band feature. Additionally, the reference to an “M” shape in Figure 3j is also vague and some guiding annotations would help.
4. Spectral Weight Contributions Analysis: In the same paragraph, the authors point out “spectral weight contributions from monolayer and bilayer graphene can also be selectively enhanced by reversing the bias voltage, as indicated by red arrow in Fig. 3j and blue arrow in Fig. 3a.” This statement is not immediately evident from the 2D color maps and needs to be substantiated with quantitative linecuts. In the linecuts in Fig. 3k, the contrast of “1ML peak doesn't seem to change from -20V to 30V and the reduction of 2ML peak could be due to band structure change instead of the spectral weight for 2ML changing. Furthermore just looking at the 2D spectra in Fig. 3a-j, the intensity of the left branch, which is attributed to 2ML, doesn't really change much.
5. On page 10 in the first paragraph, the authors say “...in the theoretical calculations in Fig. 3l-p, where a stronger monolayer (bilayer) contribution is observed in the valence band for positive (negative) bias voltage”, again because the presented 2D dispersion, while it's clear that 1ML spectral weight become stronger (redder), the 2ML branch doesn't seem to change visually. Also, it's unclear what the authors mean by “overall similar behavior” by conduction and valence band dispersion at opposite voltages. This should be elaborated.
6. The section of “Origin of the field-tunable dichotomic electronic structure” discusses the dichotomy of bands similar to those of tBLG or tDBG. This is a different dichotomy from 1ML and 2ML band dichotomy and is an important point that the paper wants to stress. However this section relies on theoretical calculations without experimental data validating this.
7. In the same section, the authors show “energy contour projected onto each constitute graphene layer”. Since the band structure is a result of hybridization of all the graphene layers, what do the authors mean by and how do they obtain projected contour in each layer? More detailed methodology of this approach would be beneficial.
8. Theory Calculation Details: theory calculation is a critical aspect of the paper, yet the methodology is only broadly described in the supplementary materials and lacks details. A more thorough description should be provided. Especially it looks like the band structures calculated in this paper look different from those shown in previous experimental studies on the same system such as Nature volume 588, pages 66–70 (2020) and Nature Physics volume 17, pages 374–380 (2021).
9. In the supplementary material “ARPES measurements” it's mentioned that the energy resolution is 50meV but the main text reports a flat band bandwidth of 70meV with an uncertainty of 10meV? How can the uncertainty be smaller than the resolution?

10. Can the authors provide more details on the AFM measurements that determined the twist angle? For example, was this using conductive AFM mode?

In summary, while the study provides valuable insights into the electronic structure of twisted monolayer-bilayer graphene, clarity and additional data are necessary to reinforce the conclusions. The details of the calculation should be provided. I look forward to the authors' response to these points to strengthen the manuscript.

Reviewer #3 (Remarks to the Author):

The manuscript "Observation of dichotomic field-tunable electronic structure in twisted monolayer-bilayer graphene" by Zhang et al. investigates the origin of an asymmetric field-tunable phase behaviour of twisted mono-bilayer graphene (tMBG) from an electronic structure point of view. Technically, these measurements are realised by a combination of in-operando gated 2.2° tMBG and nanoARPES capabilities. The electronic structure measurements show clear asymmetries for positive and negative bias voltages, and different spectral weight contributions from monolayer and bilayer graphene are identified. From transport measurements it is known that the two phases should resemble the electronic structure of tBLG and tDBG. Combined with real-space tight-binding calculations, it is shown that the electronic structures for positive and negative bias show similarities to tBLG and tDBG, respectively. The origin of the asymmetric field-tunable effect is also traced to bias-dependent spectral weight contributions of the three layers in tMBG.

The topic of the manuscript is certainly interesting and the experiments shed light on the asymmetric field-tunable properties of tMBG. Also, the combination of gated micron-sized twisted samples with nanoARPES is state-of-the-art, and the sample system is well selected to show the capabilities of this method. The article is well written and I believe that the manuscript is well-suited to be published in Nature Communications. Here are my comments:

1) Unfortunately, some of the features described in the text are very difficult to recognise and follow in the electronic structure data. Some differences in intensity/dispersion are subtle and I do not necessarily have a good solution at hand, but I wanted to give the authors a list of points that were hard to follow, which might help to improve the manuscript:

- When I first read the manuscript, I understood that the 1ML and 2ML traits should be stronger or weaker for positive/negative bias relative to each other. This is not the case, the 2ML is always

weaker, only the contributions of the 1ML and 2ML are stronger/weaker relative to themselves, and the buried 2ML features are always weaker than the 1ML. I think this could be written more clearly and would help the reader.

- In Fig. 2a, I cannot see any replica signature at the moiré superlattice BZ corners in my printout.
- In Fig. 2i, the brown dashed lines are not visible at all, and the blue dashed lines are partially buried in the blue plotted intensity of the photoemission yield.
- In Fig. 3l, the difference in spectral weight between VB and CB, indicated by the lightly shaded blue arrow, is very small and difficult to see.
- In Fig. 3l vs. Fig. 3p, I can see that the red 1ML bands are weaker and stronger, but this is not the case (at least not readily visible) for the blue 2ML bands, contrary to what is written in the text.
- In Fig. 3k, the increase in spectral weight for the 2ML is visible, but not the increase for the 1ML.
- In Figs 4g-l the different contributions of the layers are very difficult to follow. Perhaps there is some way of making the weaker contributions clearer? What do the dotted rectangles mean?

2) In the manuscript text, it would be helpful if the authors were more explicit about the visible difference, perhaps even quantitatively in the main text. For example, on page 12 they write "... is still similar to that of tBLG due to the much weaker contribution from L1". I do not agree that the contribution of L1 is "much weaker", and this makes it very difficult to follow the text. I would rather say that although the asymmetry in the spectral weight contributions of the layers is small (barely visible!), it is still significant enough to explain the origin of the overall asymmetric behaviour seen in the tMBG.

3) In the introduction the authors refer to the asymmetric field tunable phase diagram changing from a correlated phase to a ferromagnetic phase. This is not discussed at all in the later manuscript. Is the electronic structure found in experiment and theory consistent with this phase assignment? Can one learn more about the origin of these phases from the work presented here?

4) Other minor comments:

- Page 6: (Fig. 2d-e) should be (Fig. 2d-f)?
- Caption Fig. 4, g-i: A slight rearrangement of the sentence should help: "Schematic illustrations of field-induced selectively enhanced contribution under positive bias voltage from different layers of tBLG, tMBG and tDBG (indicated by dark and light shaded colors in the top panel schematic), and calculated layer-projected energy contours at..." (In the current version, it is confusing that (top panels) is placed after "under a positive bias voltage").
- Page 12: enhanced -> enhanced

We thank all reviewers for valuing the scientific merits of our work, and for providing constructive suggestions to help us improve our manuscript. In response to the reviewer's questions, we have added new results/analysis and theoretical calculations to strengthen our conclusion. We have also revised figures and statements by following the reviewer's suggestions, which has further improved the clarity of our manuscript.

The main changes to the manuscript are listed below:

1. Regarding the spectral weight transfer of 1 ML and 2 ML bands raised by all reviewers, we have **revised Fig. 3 and added Supplementary information Fig. S8** following the reviewers' suggestions. In addition, we have also revised the discussions to state clearly that our conclusion is based on the relative spectral weight transfer between 1 ML and 2 ML bands upon gating, see page 5, line 66-69: *"We find that, although the bottom bilayer graphene (2 ML) always has a weaker spectral intensity compared with top monolayer (1 ML) graphene due to the finite probing depth, a positive bias voltage (induced electric field E_{ind} pointing from monolayer to bilayer) enhances the relative spectral weight contribution from 1 ML graphene..."*.
2. In response to questions from reviewer #2 and #3, we have **added more theoretical calculation details and references in the Methods and Supplementary information**. Moreover, we have also included lattice relaxation effect into calculations, which can more precisely capture the hybridization gap at the flat band edges, see page 19, line 296: *"Classical structural relaxation using LAMMPS^{38,39} has been implemented in the calculation."* In response to the reviewer's question, we have added more details into the Method to avoid possible confusion, please see page 20, line 304-306, *"To simulate the intensity of the ARPES data, we calculate the sublattice-resolved spectral functions using a band-unfolding method⁴⁰⁻⁴², which enables to obtain the layer- or sublattice-resolved electronic structure calculation."* We also added more calculation details into the Supplementary information, please see pages 17-18 lines 135-145: *"To simulate the intensity of the ARPES data, we calculate the sublattice-resolved spectral functions using a band-unfolding method¹⁴⁻¹⁶ based on the one-particle Green's function of the moiré supercell (SC)... This band unfolding method effectively obtains the weights for each crystal momentum vector k_{SC} from the supercell, matching the vector k from the reference unit cell through $k = k_{SC} + G_{SC}$ (here G_{SC} is the supercell reciprocal lattice vector). We repeat this unfolding approach for each of the atomic orbitals of each atom in the moiré supercell, thus obtaining the spectral weight for each sublattice in each layer, hence allowing us to plot layer and sublattice resolved contributions."*
3. We have improved the clarification of our figures and statements.
 - (1) Following reviewer #1's suggestion, we have revised the introduction to convey more proper message of our scientific merits, see page 3, line 58-61: *"Such field-tunable correlated phenomena suggest a strong modification of the electronic structure under the application of an electric field. Directly probing how the actual electronic structure evolves with electric field is therefore critical for providing a better understanding of such dichotomic field-tunable physics."*

- (2) We have added related discussion and *Fig. S11 in the supplementary information* to show the ARPES experimental geometry.
 - (3) We have *adjusted the colorscale of Fig. 2a-c* to show the replica bands better, and *revised Fig. 3 and Fig. 4* by adding guiding annotations as suggested by reviewer #2 and #3.
 - (4) We have added the comparison between ARPES data with calculation results into *supplementary information Fig. S7* to support the field-tunable flat band dispersion.
 - (5) We have corrected a few typos and notations following the reviewers' suggestions.
4. We have moved previous Extended Data Figures into the Supplementary information figures to conform with the format of Nature communications.

Below is a detailed point-by-point response to the reviewers' comments.

Reviewer #1 (Remarks to the Author):

In the submitted paper, Zhang and co-authors study the angle-resolved photoemission spectra of single-gated twisted monolayer-bilayer graphene (tMBG). They show that at small twist angles ~ 2 degrees, gating the system leads to spectroscopic features resembling twisted bilayer graphene (tBG) or twisted double bilayer graphene (tDBG) for positive and negative bias, respectively.

The text is well written, the ARPES data is of high quality and the theoretical modelling is useful in establishing the message of the paper. However, the tunability of tMBG and its similarity to tBG/tDBG have been noticed elsewhere [Nature Physics 17, 374 (2021)]. The novelty of the submitted paper is the theoretical analysis of the wave functions correlated with the photoemission data. Given that the referenced paper already discusses the layer-resolved electronic density of states, its behavior in the external electric field and comparison to tBG/tDBG, I do not agree with the authors that the mechanism responsible for the tunability of tMBG remains elusive and am not convinced that the momentum resolution provided by ARPES generates enough novelty on its own to warrant publication in Nature Communications. This is also because, in my opinion, the comparison between ARPES and theory was not properly exploited. More detailed comments are below:

Reply: We thank the reviewer for appreciating the scientific merits and quality of our work, and for raising suggestions to help us improve the manuscript.

We agree with the reviewer that “the mechanism responsible for the tunability of tMBG remains elusive” is inaccurate, and we have revised it accordingly, see page 3, line 58-61: *“Such field-tunable correlated phenomena suggest a strong modification of the electronic structure under the application of an electric field. Directly probing how the actual electronic structure evolves with electric field is therefore critical for providing a better understanding of such dichotomic field-tunable physics.”*

Regarding the scientific merit of our work, we would like to point out that while layer-resolved electronic density of states in the real space has been deduced from the transport data and further supported by the calculations in the previous report (Nature Physics 17, 374 (2021)), our work provides a direct experimental verification of the dichotomic field-tunable electronic structure.

By combining experimental electronic structure with theoretical calculations, our work provides more insights from the electronic structure perspective. This scientific merit has also been particularly appreciated by reviewer #2 as “*the study provides valuable insights into the electronic structure of twisted monolayer-bilayer graphene*”; and by reviewer #3 as “*The topic of the manuscript is certainly interesting and the experiments shed light on the asymmetric field-tunable properties of tMBG.*”

1) In the abstract, “combing” should be “combining”.

Reply: We thank the reviewer for pointing out. This has been corrected in the revised version.

2) Fig. 1: “the measurement direction is along the BZ corner (...)” should be probably “the measurement direction is that connecting the BZ corners (...)”

Reply: We thank the reviewer for the kind suggestion, and we have revised the caption of Fig. 1 accordingly.

3) Fig. 2: in the description of panels g,h, no explanation is given of what the calculation result actually shows. Neither is it mentioned in the main text at this point that the theory shows wave function projections.

Reply: We thank the reviewer for pointing out this, and we have added the explanation of Fig. 2g-h in the figure caption, as “*Red and blue colors in h represent the projected contributions from the top 1 ML and bottom 2 ML graphene layers, respectively.*” We also discussed these panels in the main text. Please see page 6, lines 92-95 as “*Similar features are also captured by the calculated spectrum shown in Fig. 2h using an effective tight-binding model (see Method for more details), where red and blue colors represent projected contributions from the top 1 ML and bottom 2 ML graphene layers respectively.*”

4) Fig. 3, panels l-p: caption states that what is shown are the electronic dispersions. However, dispersions only provide the positions of the bands – what do the colors from blue to red reflect (again, projections of the wave functions are in fact shown)? In general, it seems that the authors use the word dispersion somewhat ambiguously. I think the experimental graphs show photoemission intensity along some directions in the reciprocal space or at constant binding energies – they do not show electronic dispersion. The distinction might be more than academic given the well-known impact of electronic pseudo-spin on the measured intensity in graphene-based materials.

Reply: We appreciate the reviewer for pointing out this issue.

For more accurate statements, we have used “**dispersion image**” for photoemission intensity measured along some directions in the reciprocal space (Fig. 1e-g, Fig. 2g, i, Fig. 3a-j), “**ARPES intensity maps**” for photoemission intensity measured at fixed binding energies (Fig. 2a-f).

For the calculated results, we use “**calculated spectrum**” for the calculated single particle spectral function along certain directions in the momentum space (Fig. 2h, j and Fig. 3l-p), and “**calculated intensity maps**” for calculated single particle spectral function at fixed energies (Fig. 4).

We agree with the reviewer that our calculated spectrum contains the information of both the dispersion and spectral weight from 1 ML and 2 ML graphene as indicated with the color code of red and blue, respectively. The corresponding figure caption has been added to Fig. 2h. Please see page 7 as *“Red and blue colors in h represent projected contributions from the top 1 ML and bottom 2 ML graphene layers, respectively.”*

Regarding the effect of the electronic pseudo-spin, we would like to point out that while pseudo-spin can modulate the ARPES intensity at different momentum positions (PRL 107, 166803 (2011); PRB 84, 125422 (2011)) due to the dipole matrix effect (Rev. Mod. Phys. 75, 473 (2003)), this does not influence our interpretation of the ARPES data in the current study. This is due to the fact that our conclusion is based on the *relative* change of the spectral weight *before and after gating* while keeping the same experimental geometry (same dipole matrix element).

5) Given that the authors discuss the changes in the spectral function highlighting charge transfer between layers and either twisted bilayer- or twisted double bilayer-like behavior, I wonder how the electron escape depth effect interplays with the reported findings.

Reply: The reviewer raised an important question about the electron escape depth effect (or ARPES probing depth), and particularly whether this might affect our conclusion about spectral weight transfer. According to the so-called “universal curve” (Fig. R1a), a photon energy of 100 eV (Fermi energy of photoelectron is 95.8 eV) corresponds to a probing depth of 5 Å, which allows to probe both the 1 ML and 2 ML graphene yet with different intensity --- the top 1 ML graphene has a stronger ARPES intensity than the bottom 2 ML graphene. Although such intensity modulation makes it difficult to extract the absolute spectral weight, our data reveal a distinct *change of the relative spectral weight contribution between 1 ML and 2 ML upon gating* (Fig. R1b-d). Namely, our conclusion is based on the change in the relative spectral weight contribution upon gating, which does not depend on the probing depth.

In response to the reviewer’s question, we have revised the manuscript to clarify that our conclusion is based on the change of the relative spectral weight. Please see page 5, line 66-69: *“We find that, although the bottom bilayer graphene (2 ML) always has weaker spectral intensity compared with top monolayer (1 ML) graphene due to the finite probing depth, a positive bias voltage (induced electric field E_{ind} pointing from monolayer to bilayer) enhances the relative spectral weight contribution from 1 ML graphene...”*. In addition, to make it easier to visualize the relative spectral weight transfer, we have added a panel Fig. 3q to Fig. 3 (see attached Fig. R2 below) to show a comparison of the momentum distribution curves (MDCs) at 30 V and -20 V.

Figure R1: Observation of relative spectral weight transfer upon gating, which does not depend on the probing depth. (a) The universal curve of the photoelectrons' inelastic mean free path as a function of the kinetic energy (Surf. Interf. Anal. 1, 2–11 (1979); Figure adapted from Nat. Rev. Methods Primers 2, 54 (2022)). (b-d) Dispersion images measured under gating voltages of 20 V, 0 V and -20 V, respectively. Although the 2 ML always have weaker intensity due to finite probing depth, the relative spectral weight transfer is observed as indicated by red and blue arrows.

Figure R2: Dichotomic field response of the electronic structure under negative and positive bias voltages in a 2.2° tMBG. (a-j), Dispersion images measured through K_1 and K_2 with bias voltages from -20 V to 30 V. Dotted curves in (a-j) are guiding curves. (k), Representative MDCs extracted at energies indicated by gray dashed arrows in (a, c, e, g, h, j) to reveal the selectively enhanced spectral weight from 1 ML (pointed by red dashed arrow) and 2 ML graphene (pointed by blue dashed arrow) under positive and negative bias voltages, respectively. (l-p), Calculated spectra at negative (l, m), zero (n), and positive (o, p) bias voltages, with red and blue colors representing the contribution from 1 ML and 2 ML graphene respectively. The values of the interlayer potential difference (Δ) used for the calculations are $\Delta = -100, -50, 0, 50$ and 100 meV respectively. (q), Comparison of MDCs extracted at -20 V and 30 V, which shows relatively enhanced 2 ML or 1 ML bands at negative and positive bias voltages.

6) Similarly to the previous comment, there is likely some impact of the electronic pseudo-spin. Taking into account this and the previous point would allow for a better test of the theoretical wave functions.

Reply: As explained in the previous response, our conclusion is *based on the change in the relative contributions from 1 ML and 2 ML graphene upon gating*, which does not depend on the absolute intensity or the electronic pseudo-spin effect.

7) I found no mention of the ARPES geometry (especially light polarization used) in the Methods – overall, a short comment might be useful for specialists.

Reply: We thank the reviewer for this kind suggestion. We have added the ARPES geometry into the **Methods** (at page 19, lines 291-292) as *“The incident light is set to p-polarized and the analyzer slit direction is horizontal (see Fig. S11 in the Supplementary information).”* We have also added Fig. R3 into the **Supplementary Information** as Fig. S11, where the photoemission plane (light red plane) and light polarization (red arrow) are labeled.

Figure R3: ARPES experimental geometry. The incident light polarization is linear and horizontal (p-polarized) as indicated by the red arrow. The right panel represents the Brillouin zone with the red line indicating the measurement direction.

Reviewer #2 (Remarks to the Author):

This manuscript presents a study of the electronic band structure in twisted monolayer-bilayer graphene with a 2.2-degree twist angle using nano-ARPES. The device has a back gate that can change both the carrier density and electric field. By cross-checking with their band structure calculation, they assign the two branches of bands observed in measurements to mainly coming from the monolayer and bilayer graphene respectively. They also suggest that the change of the spectral weight of the two branches under different gate voltages is consistent with their calculated bands under different electric fields. Finally, they point out through band structure calculation that the bands of twisted mono-bilayer graphene resemble those of twisted bilayer graphene (tBLG) or twisted double layer graphene (tDBG) when gate voltage is such that electrons are pushed more toward the monolayer graphene or the bilayer graphene. Upon reviewing the manuscript, I have several questions that I hope the authors can clarify:

Reply: We thank the reviewer for this nice summary of our work, and also for further raising constructive suggestions to help us improve our manuscript.

1. Determination of k-Space Orientation (Figures 1 and 2): From the ARPES measurement one can measure the relative momentum. How did the authors determine the absolute orientation and position in k-space? It seems that the authors attributed the bright spot in the equal energy cut at $E - E_F = 0$ in Fig.2a to monolayer graphene Dirac cone and therefore it's K1, the mini-Brillouin zone (mBZ) corner. This seems reasonable. However the basis for determining K2 is not apparent because data there is a purple blob and there's no distinct features like K1. Additionally, how is k-space orientation determine as indicated by the blue and red dashed line in Fig.2 a-f? This may require looking at the bands across the larger k-space to display original Brillouin zone (BZ) of graphene and reveal hexagons of Dirac points, such as presented in supplementary Fig.4 of the paper "Nature Physics volume 17, pages184–188 (2021)". Do the authors have data in larger momentum range to show this? And it would be nice to show the hexagons for bilayer graphene too.

Reply: We would like to point out that while a larger momentum space map covering two Brillouin zone corners is often used at the beginning of ARPES measurements to have a rough alignment of the sample (finding the Γ point), for twisted bilayers, more precise determination of the k-space orientation can be obtained once K_t , K_b from the top and bottom layers are determined. Figure R4a shows the schematic Brillouin zones (red and blue lines) and moiré superlattice Brillouin zones (gray lines). It is clear that once K_t , K_b are determined, Γ point can be determined accordingly by utilizing the two conditions: (1) Γ is on the perpendicular bisector between K_t and K_b ; (2) the distance between Γ and K_t (K_b) is 1.7 \AA^{-1} .

The advantage of our work is that the NanoARPES data are of high quality, where the K_b , K_t points can be identified in the Fermi surface maps (Fig. R4b-d). Although the bottom 2 ML graphene shows a weaker intensity compared to the top 1 ML graphene, the position of K_b can still be determined from the center of mass in the purple blob on the left (see Fig. R4b-d below). The determination of K_b , K_t in the momentum space map allows us to align the k-space orientation accurately.

Figure R4: Determination of absolute orientation from ARPES measurement. (a) Schematic drawing to show the Brillouin zone of top graphene (red), bottom (blue) graphene and moiré superlattice (gray dotted hexagons). The measurement direction (slit direction) is indicated by the black broken line. (b-d) ARPES intensity maps measured at the Fermi energy for twist angle of 3.0° , 2.6° and 2.2° .

2. Isolated Flat Band Discussion (Page 7): The manuscript states the observation of an isolated flat band. However, although it looks isolated in Fig. 2g and h with a hybridization gap, the same bright spot along a different cut in Fig. 2i and j is not isolated, but continuously connecting to the “1 ML” dispersive band. Isn't this contradictory to having an isolated flat band?

Reply: To reconcile this question, we have carried out a careful comparison between the theoretical calculations and experimental results as explained below.

1) We have performed extended theoretical calculations where lattice relaxation is also included to reveal the fine band structure. The calculated spectral weight in Fig. R5a shows gap openings at two edges of the flat band for 2.2° tMBG, which has a fixed momentum separation of $3k_m$ for a given twist angle, where k_m is the momentum separation between K_t and K_b .

2) While such tiny gap is smeared out in the experimental data due to the finite resolution, the experimental dispersion image in Fig. R5b shows a sudden change of intensity at the same momentum position where a gap opening is expected (indicated by the black dotted circle in Fig. R5b). Therefore, although the tiny gap is not directly observed from the ARPES results, the sudden change of intensity at the gap position suggests that the experimental electronic structure is in overall agreement with the theoretical calculations.

Figure R5: Determination of the flat band edges and gap position by combining calculations with ARPES data. (a) Calculated spectra along K_t - K_b direction, where tiny gaps open at the edges of the flat band as indicated by the red arrow. (b) Dispersion image measured along K_t - K_b direction, which shows a sudden intensity change at the expected gap position as indicated by the black dotted circle.

In response to the reviewer's question, we have added related discussion to clarify this, please see page 8, line 108-111: *“We note that while the hybridization gap (pointed by black arrow in Fig. 2j) is too small to be resolved from the experimental data directly, however, a sudden change of intensity is observed at the flat band edge (Fig. 2i), suggesting an overall agreement with the calculated spectrum in Fig. 2j.”*

3. Clarity of some Descriptions (Page 8): The manuscript says “For negative bias voltage, the flat band becomes more extended in the momentum space (see Fig. 3a)”. However this flat band extension is ambiguous because Fig. 3a only shows that the bright spot disappeared and there's no clear flat band feature. Additionally, the reference to an “M” shape in Figure 3j is also vague and some guiding annotations would help.

Reply: We thank the reviewer for the suggestion about adding guiding annotations. In response to the reviewer's suggestions, we have added red and blue dotted guiding curves in Fig. 3a-j, red and black guiding lines in Fig. 3l, p to point out the features of extended/dispersive band at negative/positive bias voltages, as shown in Fig. R6 attached below.

Figure R6: Dichotomic field response of the electronic structure under negative and positive bias voltages in a 2.2° tMBG. (a-j), Dispersion images measured through K_1 and K_2 with bias voltages from -20 V to 30 V. Dotted curves in (a-j) are guiding curves. (k), Representative MDCs extracted at energies indicated by gray dashed arrows in (a, c, e, g, h, j) to reveal the selectively enhanced spectral weight from 1 ML (pointed by red dashed arrow) and 2 ML graphene (pointed by blue dashed arrow) under positive and negative bias voltages, respectively. (l-p), Calculated spectra at negative (l, m), zero (n), and positive (o, p) bias voltages, with red and blue colors representing the contribution from 1 ML and 2 ML graphene. The values of the interlayer potential difference Δ used for the calculations are $\Delta = -100, -50, 0, 50$ and 100 meV respectively. (q) Comparison of MDCs extracted at -20 V and 30 V, which shows relatively enhanced 2 ML or 1 ML bands at negative and positive bias voltages.

Figure R7: Flutter versus dispersive flat band under negative or positive gating voltages. (a-e) Calculated spectra under interlayer potential of -100, -50, 0, 50 and 100 meV. (f-j) High-quality

Dispersion images measured along K_1 - K_2 direction under bias voltages of -20, -10, 0, 10 and 20 V. Calculated spectra are over-plotted onto (f-j) to have a better comparison, which show a good agreement between experimental results and calculations.

It is important to note that the evolution of the flat band can be better visualized by comparing the experimental results with theoretical calculations (Fig. R7). First, from theoretical calculations, it is shown that for the $\Delta = -100$ meV (here Δ is used to take into account the interlayer potential difference between neighboring layers under an applied bias voltage), the band near E_F from 1 ML graphene (guided by red dashed line in Fig. R7a) is flatter than that for $\Delta = 100$ meV (guided by black dashed line in Fig. R7e). Secondly, when overplotting theoretical calculations with our experimental data (Fig. R7f-j), there is an overall agreement between experimental results and theoretical calculations. To further illustrate the evolution of the band, we have added Fig. R7 to supplementary information Fig. S7.

4. Spectral Weight Contributions Analysis: In the same paragraph, the authors point out “spectral weight contributions from monolayer and bilayer graphene can also be selectively enhanced by reversing the bias voltage, as indicated by red arrow in Fig. 3j and blue arrow in Fig. 3a.” This statement is not immediately evident from the 2D color maps and needs to be substantiated with quantitative linecuts. In the linecuts in Fig.3k, the contrast of 1ML peak doesn’t seem to change from -20V to 30V and the reduction of 2ML peak could be due to band structure change instead of the spectral weight for 2ML changing. Furthermore just looking at the 2D spectra in Fig.3a-j, the intensity of the left branch, which is attributed to 2ML, doesn’t really change much.

Reply: We thank the reviewer for point out this. Following the reviewer’s suggestion, we have added a comparison of MDC linecuts measured at -20 V and 30 V biased voltages into Fig. 3q (also shown above in Fig. R6q). It is clear from -20 V to 30 V, the contribution from 2 ML graphene decreases, while the contribution from 1 ML graphene increases. Related discussions and revisions have been added to page 10, line 129-138, as “... *the relative spectral weight contributions from monolayer and bilayer graphene can also be selectively enhanced by reversing the bias voltage, as indicated by red arrow in Fig. 3j and blue arrow in Fig. 3a (also see Fig. S8 in the Supplementary information). This is also evident in the momentum distribution curves (MDCs) measured at a few representative bias voltages in Fig. 3k, where the red and blue dashed arrows indicate the relative spectral weight transfer between monolayer and bilayer graphene under positive and negative bias voltages respectively. Figure 3q shows a comparison between MDCs measured at bias voltages of -20 V (blue curve) and 30 V (red curve), where a relative spectral weight transfer from 2 ML to 1 ML bands is clearly observed when increasing the bias voltage.*”

5. On page 10 in the first paragraph, the authors say “...in the theoretical calculations in Fig. 3l-p, where a stronger monolayer (bilayer) contribution is observed in the valence band for positive (negative) bias voltage”, again because the presented 2D dispersion, while it’s clear that 1ML spectral weight become stronger (redder), the 2ML branch doesn’t seem to change visually. Also, it’s unclear what the authors mean by “overall similar behavior” by conduction and valence band dispersion at opposite voltages. This should be elaborated.

Reply: We thank the reviewer for pointing out this, and accordingly we have improved our presentation. In the revised figure (also attached as Fig. R8 below), the change of the spectral weight from 1 ML and 2 ML bands can be better resolved, as indicated by blue and red colored arrows.

Figure R8: Dichotomic field response of the electronic structure under negative and positive bias voltages in a 2.2° tMBG. (a-j), Dispersion images measured through K_1 and K_2 with bias voltages from -20 V to 30 V. Dotted curves in (a-j) are guiding curves. (k), Representative MDCs extracted at energies indicated by gray dashed arrows in (a, c, e, g, h, j) to reveal the selectively enhanced spectral weight from 1 ML (pointed by red dashed arrow) and 2 ML graphene (pointed by blue dashed arrow) under positive and negative bias voltages, respectively. (l-p), Calculated spectra at negative (l, m), zero (n), and positive (o, p) bias voltages, with red and blue colors representing the contribution from 1 ML and 2 ML graphene. The values of the interlayer potential difference Δ used for the calculations are $\Delta = -100, -50, 0, 50$ and 100 meV respectively. (q) Comparison of MDCs extracted at -20 V and 30 V, which shows relatively enhanced 2 ML or 1 ML bands at negative and positive bias voltages.

We agree with the reviewer that the statement of “overall similar behavior between conduction and valence bands at opposite voltages” is confusing. In response to the reviewer’s question, we have revised the supplementary information Fig. S9 (attached below as Fig. R9) by adding labels for “valence band” in Fig. S9c and “conduction band” in Fig. S9d. Related discussions have also been revised to make it more readable, please see page 11, line 146-150: *“Moreover, the calculated spectra show that the valence band at positive bias voltage shows similar dispersion to the conduction band at negative bias voltage (see Supplementary information Fig. S9 for more details), suggesting that reversing the bias voltage can lead to a switching between the electron and hole bands in tMBG.”*

Figure R9: Switching between electron and hole bands by reversing the bias voltage. (a-c) Calculated spectra under the bias voltage of $V_g < 0$ ($\Delta = -100$ meV), $V_g = 0$ and $V_g > 0$ ($\Delta = 100$ meV), respectively. Red and blue colors representing the contribution from 1 ML and 2 ML graphene. (d) Same calculated spectrum as (a), except that the energy axis is reversed for easy comparison with (c). Comparison between (c) with (d) suggests an overall similar behavior of the electron and hole bands under opposite bias voltages.

6. The section of “Origin of the field-tunable dichotomic electronic structure” discusses the dichotomy of bands similar to those of tBLG or tDBG. This is a different dichotomy from 1ML and 2ML band dichotomy and is an important point that the paper wants to stress. However this section relies on theoretical calculations without experimental data validating this.

Reply: We thank the referee for this comment. We would like to point out that both the dispersion images and the intensity maps correspond to slices of the electronic structure E - k_x - k_y at fixed k_x or k_y (for dispersion images) or at different energies (for intensity maps), which could provide different projections of the E - k_x - k_y space to elaborate the band dispersion. The nice agreement between experimental results and theoretical calculations is confirmed by comparing the dispersion images with calculated spectra in Fig. 3, which provides strong support for the interpretation of the experimental results. We note that the intensity maps shown in Fig. 4 present the same calculated E - k_x - k_y data from a different projection.

In short, the dichotomies in the dispersion images and intensity maps are the effectively same, except that the layer-projected intensity maps can provide further insights to understand the evolution of electronic structure in tMBG between tBLG and tDBG. The comparison between our theoretical calculations still can provide more information for a deeper understanding of the field-tunable properties of tMBG.

7. In the same section, the authors show “energy contour projected onto each constitute graphene layer”. Since the band structure is a result of hybridization of all the graphene layers, what do the authors mean by and how do they obtain projected contour in each layer? More detailed methodology of this approach would be beneficial.

Reply: We thank the reviewer for pointing out this. We would like to clarify that the real space tight-binding calculations, which has been commonly applied, can give the dispersion in the moiré superlattice Brillouin zone (folded band structure), similar to the effective continuum model used in two references pointed by the reviewer (Nature 588, 66–70 (2020); Nature Physics 17, 374–380 (2021)). In order to compare our experimental data measured in the

graphene Brillouin zone with the calculated results, a band unfolding is performed, which can project the spectral weight into the primitive Brillouin zone of different graphene layers and obtain layer-resolved band structure. Such band unfolding method has been previously utilized in calculation of spectral function of moiré systems (J. Phys.: Condens. Matter 25, 345501 (2013); PRB 95, 085420 (2017)), and we have included these papers as reference in the revised manuscript.

In response to the reviewer's question, we have added more details about the calculation with related references into the Method, see page 20, line 304-306: *"To simulate the intensity of the ARPES data, we calculate the sublattice-resolved spectral functions using a band-unfolding method⁴⁰⁻⁴², which enables to obtain the layer- or sublattice-resolved electronic structure calculation."*

8. Theory Calculation Details: theory calculation is a critical aspect of the paper, yet the methodology is only broadly described in the supplementary materials and lacks details. A more thorough description should be provided. Especially it looks like the band structures calculated in this paper look different from those shown in previous experimental studies on the same system such as Nature volume 588, pages 66–70 (2020) and Nature Physics volume 17, pages 374–380 (2021).

Reply: We would like to clarify that the electronic structure in the references pointed out by the reviewer (Nature 588, 66–70 (2020); Nature Physics 17, 374–380 (2021)) were plotted in the moiré superlattice Brillouin zone, not the graphene Brillouin zone. Since the ARPES data were measured in the graphene Brillouin zone, in order to have a fair comparison between the theoretical calculation and experimental data, we have further performed the band-unfolding spectral weight calculation following a standard procedure (PRL 104, 216401 (2010); J. Phys.: Condens. Matter 25, 345501 (2013); PRB 95, 085420 (2017)), to project the spectral weight back to the graphene Brillouin zone.

In response to the reviewer's question, we have added more details into the Method to avoid possible confusion, please see page 20, lines 304-306: *"To simulate the intensity of the ARPES data, we calculate the sublattice-resolved spectral functions using a band-unfolding method⁴⁰⁻⁴², which enables us to obtain the layer- and sublattice-resolved electronic structure calculation."*

We also added more calculation details into the Supplementary information, please see pages 17-18, line 135-145: *"To simulate the intensity of the ARPES data, we calculate the sublattice-resolved spectral functions using a band-unfolding method¹⁵⁻¹⁷ based on the one-particle Green's function of the moiré supercell (SC)... This band unfolding method effectively obtains the weights for each crystal momentum vector k_{SC} from the supercell, matching the vector k from the reference unit cell through $k = k_{SC} + G_{SC}$ (here G_{SC} is the supercell reciprocal lattice vector). We repeat this unfolding approach for each of the atomic orbitals of each atom in the moiré supercell, thus obtaining the spectral weight for each sublattice in each layer, hence allowing us to plot layer- and sublattice-resolved contributions."*

9. In the supplementary material “ARPES measurements” it’s mentioned that the energy resolution is 50meV but the main text reports a flat band bandwidth of 70 meV with an uncertainty of 10 meV? How can the uncertainty be smaller than the resolution?

Reply: We note that the finite energy resolution will lead to broadening of the measured dispersion, however, a change of 10 meV can still be resolved through curve fitting. Figure R10 shows two simulated energy distribution curves (EDCs) with a full-width at half maximum (FWHM) of 50 meV and an energy difference of 10 meV. The peak position difference is clearly resolved, suggesting that the energy resolution is sufficient for resolving the bandwidth of flat band with an uncertainty of 10 meV.

Figure R10: Simulated EDCs to check if it is possible to resolve peaks with an energy difference of 10 meV. The FWHM of a single Lorentz peak (red and blue curves) is set to 50 meV to simulate the energy resolution of the experimental results.

10. Can the authors provide more details on the AFM measurements that determined the twist angle? For example, was this using conductive AFM mode?

Reply: To determine the twist angle, we use the lateral force AFM (L-AFM) measurement. In our L-AFM measurements, we used silicon nitride probes to obtain the lateral force and topography image under the contact mode. We note that L-AFM measurement is particularly sensitive to the moiré superlattice period because the stick-slip effect would occur at the moiré superlattice scale (Phys. Rev. Lett. 128, 226101 (2022)). After extracting the moiré superlattice period λ_m from L-AFM, the precise twist angle θ can be further determined by

$$\lambda_m = \frac{a}{2 \sin \frac{\theta}{2}}$$

In response to the reviewer’s question, we have added more details about AFM measurements in the methods, see page 19, line 282-287: *“The twist angle can be determined by combining NanoARPES and lateral force AFM (L-AFM) measurements. For the L-AFM measurements, silicon nitride probes were used to obtain the lateral force and topography image under the contact mode. We note that L-AFM measurement is particularly sensitive to the moiré superlattice period because the stick-slip effect would occur at the moiré superlattice scale³⁷. After extracting the moiré superlattice period λ_m from L-AFM measurements, the twist angle θ can be further determined by $\lambda_m = a/(2\sin(\theta/2))$.”*

In summary, while **the study provides valuable insights into the electronic structure of twisted monolayer-bilayer graphene**, clarity and additional data are necessary to reinforce the conclusions. The details of the calculation should be provided. I look forward to the authors' response to these points to strengthen the manuscript.

Reply: We thank the reviewer for valuing the scientific merits of our work, and for raising these valuable comments, which have led to a significant improvement of our manuscript. With the revisions, we hope that the manuscript can now be accepted for publication.

Reviewer #3 (Remarks to the Author):

The manuscript "Observation of dichotomic field-tunable electronic structure in twisted monolayer-bilayer graphene" by Zhang et al. investigates the origin of an asymmetric field-tunable phase behaviour of twisted mono-bilayer graphene (tMBG) from an electronic structure point of view. Technically, these measurements are realized by a combination of in-operando gated 2.2° tMBG and nanoARPES capabilities. The electronic structure measurements show clear asymmetries for positive and negative bias voltages, and different spectral weight contributions from monolayer and bilayer graphene are identified. From transport measurements it is known that the two phases should resemble the electronic structure of tBLG and tDBG. Combined with real-space tight-binding calculations, it is shown that the electronic structures for positive and negative bias show similarities to tBLG and tDBG, respectively. The origin of the asymmetric field-tunable effect is also traced to bias-dependent spectral weight contributions of the three layers in tMBG. **The topic of the manuscript is certainly interesting and the experiments shed light on the asymmetric field-tunable properties of tMBG. Also, the combination of gated micron-sized twisted samples with nanoARPES is state-of-the-art, and the sample system is well selected to show the capabilities of this method. The article is well written and I believe that the manuscript is well-suited to be published in Nature Communications.** Here are my comments:

Reply: We thank the reviewer for appreciating the scientific merits of our work and recommendation for publication. We would also like to thank the reviewer for constructive suggestions to help us further improve the manuscript.

1) Unfortunately, some of the features described in the text are very difficult to recognize and follow in the electronic structure data. Some differences in intensity/dispersion are subtle and I do not necessarily have a good solution at hand, but I wanted to give the authors a list of points that were hard to follow, which might help to improve the manuscript:

- When I first read the manuscript, I understood that the 1ML and 2ML traits should be stronger or weaker for positive/negative bias relative to each other. This is not the case, the 2ML is always weaker, only the contributions of the 1ML and 2ML are stronger/weaker relative to themselves, and the buried 2ML features are always weaker than the 1ML. I think this could be written more clearly and would help the reader.

Reply: We apologize for the confusion. We would first like to clarify that our conclusion is based on the relative spectral transfer between 1 ML and 2 ML upon gating, not the *absolute* spectral weight. To avoid possible confusion, we have revised the manuscript to point to the *"relative"* contribution, see page 5, line 66-71: *"...although the bottom bilayer graphene (2 ML) always has a much weaker spectral compared to the top monolayer (1 ML) graphene, a*

positive bias voltage (induced electric field E_{ind} pointing from monolayer to bilayer) enhances the **relative** spectral weight contribution from 1 ML graphene (pointed by red arrow in Fig. 1e) as well as the conical shape, while a negative bias voltage enhances the **relative** contribution from 2 ML graphene... ”.

- In Fig. 2a, I cannot see any replica signature at the moiré superlattice BZ corners in my printout.

Reply: We agree with the reviewer that the replica signatures at the moiré superlattice BZ corners are rather weak and difficult to resolve. To make the replica pockets clearer, we have adjusted the color scale of Fig. 2a-c by using a log scale. Four weak intensity spots can be observed in the moiré superlattice BZ corners (marked by red circles). In addition, the moiré bands are more clearly resolved in the dispersion images shown in Fig. R11g, i.

In response to the reviewer’s question, we have replaced Fig. 2a-c with a better colorscale.

Figure R11: Fermi surface topology and coexistence of monolayer and bilayer graphene features in the 2.2° tMBG. (a-f), ARPES intensity maps measured at energies from E_F to -0.50 eV. The red and blue lines mark the Brillouin zone boundaries for top monolayer graphene and bottom bilayer graphene respectively. (g, h) Dispersion images measured along the black line indicated by the inset of (g), and calculated spectrum for comparison. Red and blue colors in (h) represent the projected contributions from the top 1 ML and bottom 2 ML graphene layers, respectively. Red and black arrows point to the flat band and hybridization gap respectively. (i, j) Dispersion image measured along the black line indicated by the inset of (i), and calculated spectrum for comparison.

- In Fig. 2i, the brown dashed lines are not visible at all, and the blue dashed lines are partially buried in the blue plotted intensity of the photoemission yield.

Reply: We thank the reviewer for pointing out this. The brown and blue dashed curves have been enhanced in Fig. 2i in the revised manuscript, which is also attached as Fig. R12 below.

Figure R12: Switching between electron and hole bands by reversing the bias voltage. (a-c) Calculated spectra under the bias voltage of $V_g < 0$ ($\Delta = -100$ meV), $V_g = 0$ and $V_g > 0$ ($\Delta = 100$ meV), respectively. Red and blue colors representing the contribution from 1 ML and 2 ML graphene. (d) Same calculated spectra as (a), except that the energy axis is reversed for easy comparison with (c). Comparison between (c) with (d) suggests an overall similar behavior of the electron and hole bands under opposite bias voltages.

- In Fig. 3l, the difference in spectral weight between VB and CB, indicated by the lightly shaded blue arrow, is very small and difficult to see.

Reply: We would like to clarify that the comparison of the VB (or CB) upon gating is the main focus of this work, not between the VB and CB. To make it clearer, we have adjusted the colorscale and labels in Fig. 3l-p (see Fig. R13 attached below).

- In Fig. 3l vs. Fig. 3p, I can see that the red 1ML bands are weaker and stronger, but this is not the case (at least not readily visible) for the blue 2ML bands, contrary to what is written in the text.

Reply: We agree with the reviewer that compared with the change of 1 ML spectral weight upon gating, the change for 2 ML bands is weaker. We have adjusted the colorscale in Fig. 3l-p to have a better visualization of the spectral weight change from 1 ML and 2 ML graphene (See Fig. R13 below), and added panel Fig. 3q to compare MDCs measured at -20 V and 30 V. In addition, we have also revised the discussions to make it explicit that our conclusions are based on the “*relative*” spectral weight contribution upon gating, see page 10, lines 140-143: “...where a stronger *relative spectral weight contribution* is observed for monolayer valence band at positive bias voltage. Interestingly, at negative bias voltage (Fig. 3l), the *enhanced relative spectral weight contribution* from bilayer graphene in the valence band (blue arrow in Fig. 3l) is also accompanied by a reduced spectral weight contribution...”

Figure R13: Dichotomic field response of the electronic structure under negative and positive bias voltages in a 2.2° tMBG. (a-j), Dispersion images measured through K_1 and K_2 with bias voltages from -20 V to 30 V. Dotted curves in (a-j) are guiding curves. (k), Representative MDCs extracted at energies indicated by gray dashed arrows in (a, c, e, g, h, j) to reveal the selectively enhanced spectral weight from 1 ML (pointed by red dashed arrow) and 2 ML graphene (pointed by blue dashed arrow) under positive and negative bias voltages, respectively. (l-p), Calculated spectra at negative (l, m), zero (n), and positive (o, p) bias voltages, with red and blue colors representing the contribution from 1 ML and 2 ML graphene. The values of the interlayer potential difference Δ used for the calculations are $\Delta = -100, -50, 0, 50$ and 100 meV respectively. (q) Comparison of MDCs extracted at -20 V and 30 V, which shows relatively enhanced 2 ML or 1 ML bands at negative and positive bias voltages.

- In Fig. 3k, the increase in spectral weight for the 2ML is visible, but not the increase for the 1ML.

Reply: We thank the reviewer for pointing this out. To make it easier to see the change in both 1 ML and 2 ML graphene upon gating, we have added panel Fig. 3q a comparison of MDCs measured at -20 V and 30 V respectively. It is now clear that when the biased voltage changes from -20 V to 30 V, spectral contribution from 2 ML graphene decreases (pointed by blue arrow in Fig. 3q) while spectral contribution from 1 ML graphene enhances (pointed by red arrow in Fig. 3q). Related discussions have also been added to page 10, line 132-138: *“This is also evident in the momentum distribution curves (MDCs) measured at a few representative bias voltages in Fig. 3k, where the red and blue dashed arrows indicate the relative spectral weight transfer to monolayer and bilayer graphene under positive and negative bias voltages respectively. Figure 3q shows a comparison between MDCs measured at bias voltages of -20 V (blue curve) and 30 V (red curve), where a relative spectral weight transfer from 2 ML to 1 ML bands is clearly observed when increasing the bias voltage.”*

- In Figs 4g-l the different contributions of the layers are very difficult to follow. Perhaps there is some way of making the weaker contributions clearer? What do the dotted rectangles mean?

Reply: We apologize for the confusion. In previous Fig. 4, we intended to use the solid and dotted rectangles to distinguish the similarity among tBLG and tDBG. To have a better

clarification, we have revised Fig. 4 (also shown as Fig. R14 below), by using broken rectangles to indicate the similarity to tBLG at $V_g > 0$ and to tDBG at $V_g < 0$.

Figure R14: Field-tunable electronic structure from theoretical calculations, and layer-resolved energy contours for positive and negative bias voltages. (a-f), Calculated energy contours at -0.2 eV for tBLG, tMBG and tDBG under positive (a-c) and negative (d-f) bias voltages with interlayer potential difference of $\Delta = 100$ meV and -100 meV respectively. The scale bar in (a) is 0.1\AA^{-1} . (g-i), Schematic illustrations of field-induced selectively enhanced contribution under positive bias voltage from different layers of tBLG, tMBG and tDBG (indicated by dark and light shaded colors in the top panel schematic), and calculated layer-projected energy contours at -0.2 eV (lower panels, positive bias voltage with $\Delta = 100$ meV). (j-l), Schematic illustrations of field-induced selectively enhanced contribution from different layers of tBLG, tMBG and tDBG under a negative bias voltage (top panels), and calculated layer-projected energy contours at -0.2 eV (lower panels, $\Delta = -100$ meV).

2) In the manuscript text, it would be helpful if the authors were more explicit about the visible difference, perhaps even quantitatively in the main text. For example, on page 12 they write "... is still similar to that of tBLG due to the much weaker contribution from L1". I do not agree that the contribution of L1 is "much weaker", and this makes it very difficult to follow the text. I would rather say that although the asymmetry in the spectral weight contributions of the layers is small (barely visible!), it is still significant enough to explain the origin of the overall asymmetric behaviour seen in the tMBG.

Reply: We thank the reviewer for the kind suggestion. Accordingly, we have revised the discussions of Fig. 4, please see page 13, line 174-182: *"Therefore, although tMBG has one more layer (bottom layer, L_1) than tBLG, the energy contour for tMBG under positive bias voltage is still similar to that of tBLG due to the smaller contribution of spectral weight from L_1 Although the modulation in the spectral weight contribution of different layers is small,*

the asymmetric shape of pockets under reversed bias voltage is still significant enough to explain the origin of the overall asymmetric behavior in the tMBG”.

3) In the introduction the authors refer to the asymmetric field tunable phase diagram changing from a correlated phase to a ferromagnetic phase. This is not discussed at all in the later manuscript. Is the electronic structure found in experiment and theory consistent with this phase assignment? Can one learn more about the origin of these phases from the work presented here?

Reply: We appreciate the reviewer for bringing out this essential question. The main origin for the assigned different phases is the similarity to either tBLG (correlated insulators) or tDBG (ferromagnetic states), as reported in previous work (Nat. Phys. 17, 374 (2021)), the observation of asymmetric field tunable phase diagram is explained by redistribution of LDOS among layers, which leads to the similar phases to tBLG or tDBG when switching the electric fields. Here, by performing ARPES measurements and theoretical calculations, we provide direct experimental evidence to support the proposed mechanism, namely, the field-selective enhancement of contributions from 1 ML or 2 ML bands, which induces the field-tunable electronic structure that resembles tBLG or tDBG. More importantly, by projecting the spectral contribution from each individual layer in the momentum space, our work also provides more complete and microscopic information on how the electric field actually tunes the electronic structure of each individual layer. In summary, our results provide more insights to understand the tunable electronic structure of tMBG, which can exhibit correlated phases resembles either tBLG or tDBG under opposite bias voltages, including the previous reported correlated or ferromagnetic phases.

In response to the reviewer’s suggestion, we have revised related discussions to make it more explicit, see page 14, line 186-191: *“While a field-tunable electronic structure has been deduced from the asymmetric phase diagram reported in transport measurements⁶, our work provides direct electronic structure insights for understanding the field-tunable physics. In particular, by projecting the spectral contribution from each individual layer in the momentum space, our work also provides more complete and microscopic information on how the electric field actually tunes the electronic structure of each individual layer.”*

4) Other minor comments:

- Page 6: (Fig. 2d-e) should be (Fig. 2d-f)?

Reply: We thank the reviewer for pointing this out. This has been corrected in the revised manuscript.

- Caption Fig. 4, g-i: A slight rearrangement of the sentence should help: “Schematic illustrations of field-induced selectively enhanced contribution under positive bias voltage from different layers of tBLG, tMBG and tDBG (indicated by dark and light shaded colors in the top panel schematic), and calculated layer-projected energy contours at...” (In the current version, it is confusing that (top panels) is placed after “under a positive bias voltage”).

Reply: We thank the reviewer for the suggestion, and we have revised the statement following the reviewer’s comment, see highlights in the figure caption of Fig. 4.

- Page 12: enhanced -> enhanced

Reply: We thank the reviewer for pointing out this. We have corrected the typo.

REVIEWER COMMENTS

Reviewer #2 (Remarks to the Author):

The authors have provided satisfactory responses to most of my questions. However, I still have the following two further comments.

1. About point #6 in my previous round of review and the authors' corresponding response. I understand "the dichotomies in the dispersion images and intensity maps are the effectively same, except that the layer-projected intensity maps can provide further insights to understand the evolution of electronic structure in tMBG between tBLG and tDBG ", and that's not what I have issue with. What I have issue with is that the conclusion of dichotomy of tMBG bands being similar to those of tBLG or tDBG under opposite gate voltages solely comes from theory calculation (i.e., figure 4). While I understand the logic presented in the paper seems to be that Fig. 1 to 3 have established that the calculation and the ARPES data seem consistent with each other, therefore Fig. 4, though purely calculation, should also reflect what's happening in the material. While I think this level of argument is perfectly fine, I want to point out that this is certainly different from (and weaker than) directly showing measured intensity maps like those Fig.2 but at large positive and negative gate voltages. And in the abstract the wording "Specifically, dispersive bands similar to tBLG are observed under positive bias voltage, while in contrast, more pronounced flat bands resembling twisted double bilayer graphene (tDBG) are observed under reversed bias voltage" right after "Interestingly, selective enhancement of relative spectral weight contributions from monolayer and bilayer graphene is observed when switching the polarity of the bias voltage." make it sound like both effects were directly observed in experiments but in fact only the latter was directly shown by data and the former was argued through calculation.

I think the authors should either weaken their statement (particularly in the abstract) or add two intensity maps at large opposite voltages in Fig. 4, along with the theory calculation, unless the authors can offer some new arguments.

2. About point #8. I thank the authors for providing more details on the calculation although it still seems a bit brief. For example, using tight-binding model for solving twisted graphene usually involves large number of atoms. What are some of the specific parameters that the authors used? I'm only asking this because it seems that tight-binding combined with band unfolding has not been typically used since the prominent experimental discovery in 2018 (maybe I'm ignorant and there were such theory studies). Is it possible that the authors could show some of the codes for their calculation? If not, maybe the authors can answer some of my specific questions.

- a. Can the authors show their calculated bands before the unfolding procedure so that one can compare them with the existing calculated bands in the moire superlattice Brillouin zone?
- b. In the reference PRB 95, 085420 (2017) added by the authors, the band unfolding was done at commensurate twist angle. Do the authors have ways to go around the issue of calculation at incommensurate twist angles?

Reviewer #3 (Remarks to the Author):

The authors have addressed all my concerns and questions, and have greatly improved the presentation of the figures and their discussion in the main text.

I have one last optional comment: I still think that the visibility of the features in Fig. 2 could be improved. The grey BZ in Fig. 2a is very faint and hard to see, and the blue and brown dashed lines in Fig. 1 are also hard to see.

We thank both reviewers for appreciating our efforts during the last response, and for providing further suggestions to help us improve our manuscript.

Reviewer #2 (Remarks to the Author):

The authors have provided satisfactory responses to most of my questions. However, I still have the following two further comments.

1. About point #6 in my previous round of review and the authors' corresponding response. I understand "the dichotomies in the dispersion images and intensity maps are the effectively same, except that the layer-projected intensity maps can provide further insights to understand the evolution of electronic structure in tMBG between tBLG and tDBG", and that's not what I have issue with. What I have issue with is that the conclusion of dichotomy of tMBG bands being similar to those of tBLG or tDBG under opposite gate voltages solely comes from theory calculation (i.e., figure 4). While I understand the logic presented in the paper seems to be that Fig. 1 to 3 have established that the calculation and the ARPES data seem consistent with each other, therefore Fig. 4, though purely calculation, should also reflect what's happening in the material. While I think this level of argument is perfectly fine, I want to point out that this is certainly different from (and weaker than) directly showing measured intensity maps like those Fig.2 but at large positive and negative gate voltages. And in the abstract the wording "Specifically, dispersive bands similar to tBLG are observed under positive bias voltage, while in contrast, more pronounced flat bands resembling twisted double bilayer graphene (tDBG) are observed under reversed bias voltage" right after "Interestingly, selective enhancement of relative spectral weight contributions from monolayer and bilayer graphene is observed when switching the polarity of the bias voltage." make it sound like both effects were directly observed in experiments but in fact only the latter was directly shown by data and the former was argued through calculation.

I think the authors should either weaken their statement (particularly in the abstract) or add two intensity maps at large opposite voltages in Fig. 4, along with the theory calculation, unless the authors can offer some new arguments.

Reply: We thank the reviewer for the suggestion. In response to the reviewer's suggestion, we have revised the abstract make it clearer, see page 2, lines 34-39: *"Interestingly, selective enhancement of the relative spectral weight contributions from monolayer and bilayer graphene is observed when switching the polarity of the bias voltage. Combining experimental results with theoretical calculations, the origin of such field-tunable electronic structure, resembling either tBLG or twisted double-bilayer graphene (tDBG), is attributed to the selectively enhanced contribution from different stacking graphene layers with a strong electron-hole asymmetry..."*

2. About point #8. I thank the authors for providing more details on the calculation although it still seems a bit brief. For example, using tight-binding model for solving twisted graphene

usually involves large number of atoms. What are some of the specific parameters that the authors used? I'm only asking this because it seems that tight-binding combined with band unfolding has not been typically used since the prominent experimental discovery in 2018 (maybe I'm ignorant and there were such theory studies). Is it possible that the authors could show some of the codes for their calculation? If not, maybe the authors can answer some of my specific questions.

Reply: Regarding the question about the calculation, we have added related calculation details to the revised manuscript and the Supplementary information. Please see page 15, lines 295-301: *"...We use the effective tight-binding model parameters for ε_i and t_{ij} using either the monolayer³⁸ version or the Bernal bilayer³⁹ version depending on which part of the system is under consideration. We set the effective nearest hopping term $t_0 = -3.1$ eV and the interlayer hopping scaling factor $S = 0.895$ for the Scaled Hybrid Exponential (SHE) model⁴⁰, to effectively match the Fermi velocity $v_F \approx 1 \times 10^6$ m/s and the magic angle 1.08° of tBLG. In the calculations, a commensurate superlattice is used, and here we use a twist angle of 2.28° to perform the calculations, which is the closest commensurate twist angle near 2.2° ..."* in the revised manuscript, and see page 16, lines 117-122: *"...We use the effective tight-binding model parameters for ε_i and t_{ij} using either the monolayer⁶ version or the Bernal bilayer⁷ version depending on which part of the system is under consideration. We set the effective nearest hopping term $t_0 = -3.1$ eV and the interlayer hopping scaling factor $S = 0.895$ for the Scaled Hybrid Exponential (SHE) model⁸, to effectively match the Fermi velocity $v_F \approx 1 \times 10^6$ m/s and the magic angle 1.08° of tBLG. In the calculations, a commensurate superlattice is used, and here we use a twist angle of 2.28° to perform the calculations, which is the closest commensurate twist angle near 2.2° ..."* in the supplementary information.

The band unfolding is realized by using the Wannier functions (*Phys. Rev. Lett.* 104, 216401 (2010); *Phys. Rev. B* 106, 115410 (2022)), and the main code can be found as "WannierTools" (*Comput. Phys. Commun.* 224: 405-416 (2018)) in the "Github" website, which is open-source software for calculating the physical properties of given tight-binding models. Since the band unfolding is well-established method in the calculations of twisted graphene systems, we have included the above references into the manuscript, which provides necessary information for repeating our calculation results, meanwhile keeping the manuscript concise. See page 16, lines 312-313: *"...The band unfolding code can be found as WannierTools⁴⁶ on the GitHub. See Fig. S11 and more calculation details in the Supplementary Information."* in the revised manuscript. We have also added the related references and discussions to the Supplementary information, see page 18 of Supplementary information, lines 147-149, *"The main code we used is called the WannierTools¹⁹, which is an open source software in the GitHub that studies the physical properties of a given tight-binding model."*

a. Can the authors show their calculated bands before the unfolding procedure so that one can compare them with the existing calculated bands in the moiré superlattice Brillouin zone?

Reply: Following the reviewer's suggestion, we show in Fig. R1 below the calculated bands before and after the band unfolding procedure for a better comparison. By performing the band unfolding procedure, the calculated band structure (Fig. R2a) is projected into the graphene

Brillouin zone (Fig. R2b), which can be better compared with the ARPES data. We have also added such comparison as Fig. R2 as Fig. S11 in the Supplementary information, and added related discussions on page 18, lines 149-152: “Figure S11 shows a comparison between the calculated band structure before (Fig. S11a) and after band unfolding onto the top graphene Brillouin zone (Fig. S11b), which shows overall agreement, meanwhile unfolded calculation results (Fig. S11b) can be better compared with the experimental results”.

Figure R1: Comparison between the calculations before and after the band unfolding. (a) Calculated band structure in the moiré Brillouin zone before band unfolding. (b) Calculated band structure after band unfolding procedure. (c) Comparison between the calculations before and after band unfolding by directly folding (a) onto (b). The k -path for panels (a-c) is the same, by cutting through both the top (K_t) and bottom (K_b) K points

b. In the reference PRB 95, 085420 (2017) added by the authors, the band unfolding was done at commensurate twist angle. Do the authors have ways to go around the issue of calculation at incommensurate twist angles?

Reply: We are sorry for the confusion, and we would like to clarify here our calculation is based on 2.28° , which corresponds to a commensurate structure near 2.2° . The well-defined crystal momenta in the supercell is the prerequisite for performing band unfolding calculations. For incommensurate twist angles, there is no repeatable unit and the reciprocal lattice is not well-defined. Therefore, we built the commensurate twist graphene systems at 2.28° near 2.2° to perform the tight-binding model calculations and band unfolding. We have revised the Methods part in the revised manuscript to include this information, see page 16s, lines 299-301: “...In the calculations, a commensurate superlattice is used, and here we use a twist angle of 2.28° to perform the calculations, which is the closest commensurate twist angle near 2.2° ”.

Reviewer #3 (Remarks to the Author):

The authors have addressed all my concerns and questions, and have greatly improved the presentation of the figures and their discussion in the main text.

I have one last optional comment: I still think that the visibility of the features in Fig. 2 could be improved. The grey BZ in Fig. 2a is very faint and hard to see, and the blue and brown

dashed lines in Fig. I are also hard to see.

Reply: We thank the reviewer for the helpful suggestions. We have modified Fig. 2a, i to improve the visibility, as attached below in Fig. R2.

Figure R2: Fermi surface topology and coexistence of monolayer and bilayer graphene features in the 2.2° tMBG. (a-f) ARPES intensity maps measured at energies from E_F to -0.50 eV. The red and blue lines mark the Brillouin zone boundaries for top monolayer graphene and bottom bilayer graphene respectively. (g,h) Dispersion image measured along the black line indicated by the inset of (g) and calculated spectrum for comparison. Red and blue colors in (h) represent projected contributions from the top 1 ML and bottom 2 ML graphene layers, respectively. Red and black arrows point to the flat band and hybridization gap respectively. (i,j) Dispersion image measured along the black line indicated by the inset of (i), and the calculated spectrum for comparison.